# Influence of El Niño on the variability of global shoreline position

Rafael Almar [1] ✉, Julien Boucharel [1,2] ✉, Marcan Graffin[1], Gregoire Ondoa Abessolo [3], Gregoire Thoumyre [1], Fabrice Papa [1,4], Roshanka Ranasinghe [5,6,7], Jennifer Montano[8], Erwin W. J. Bergsma [9], Mohamed Wassim Baba[10] & Fei-Fei Jin [2]

Coastal zones are fragile and complex dynamical systems that are increasingly under threat from the combined effects of anthropogenic pressure and climate change. Using global satellite derived shoreline positions from 1993 to 2019 and a variety of reanalysis products, here we show that shorelines are under the influence of three main drivers: sea-level, ocean waves and river discharge. While sea level directly affects coastal mobility, waves affect both erosion/accretion and total water levels, and rivers affect coastal sediment budgets and salinity-induced water levels. By deriving a conceptual global model that accounts for the influence of dominant modes of climate variability on these drivers, we show that interannual shoreline changes are largely driven by different ENSO regimes and their complex inter-basin teleconnections. Our results provide a new framework for understanding and predicting climate-induced coastal hazards.

Coastal areas host a considerable part of human life and activities, providing tremendous societal, economic and ecological benefits. The health of these ecosystems, however, highly depends on the fragile balance between climate influence and local anthropogenic constraints[1,2]. Coastal erosion and flooding, associated with land use changes have already placed seafront ecosystems and population at great risk[3–8] and this is only expected to worsen in the future[9]. On a decadal to centennial time scale, sea level rise and river influences will dominate, compared to waves, which are expected to show more contrasting trends globally[10]. Therefore, understanding and predicting shoreline evolution is of great importance for coastal zone management, to anticipate potential threats well in advance, such that there is sufficient lead-time to implement effective adaptation measures[11,12].

However, it remains extremely challenging to predict medium (e.g. seasonal and inter-annual) to long- (decadal to century) term shoreline evolution due to the intrinsic limitations of current means of coastal observation and coastal research approaches[1,13–15]. One of the main obstacles impairing a worldwide assessment of coastal morphological change originates from the lack of long-term observational data at a global scale. The advent of earth observation from space has greatly increased the availability of optical satellite data at global scale[16,17], which in combination with the computational power offered by cloud-based platforms[18,19], have recently enabled global scale assessments of shoreline evolution[20,21] over the past three decades or so. Within this line of research, the linkages between observed shoreline changes and their potential dynamic drivers have yet to be analyzed in a comprehensive way to provide reliable projections of how the world's shorelines may evolve in response to climate change[22–24]. A comprehensive physics-based approach has yet to be developed globally. This is currently impossible due to the different scales at play

[1]LEGOS (Université de Toulouse/CNRS/IRD/UPS), Toulouse, France. [2]Department of atmospheric sciences (University of Hawaii at Manoa), Honolulu, USA. [3]Ecosystems and Fishery Resources Laboratory, Institute of Fisheries and Aquatic Sciences, University of Douala, Douala, Cameroon. [4]Universidade de Brasília (UnB), IRD, Instituto de Geociencias, Brasilia, Brazil. [5]Department of Coastal and Urban Risk & Resilience, IHE Delft Institute for Water Education, P.O. Box 3015, 2610 DA Delft, The Netherlands. [6]Harbour. Coastal and Offshore Engineering, Deltares, PO Box 177, 2600 MH Delft, The Netherlands. [7]Water Engineering and Management, Faculty of Engineering Technology, University of Twente, PO Box 217, 7500 AE Enschede, The Netherlands. [8]GET (Université de Toulouse/ CNRS/IRD/UPS), Toulouse, France. [9]Earth Observation Lab, French Space Agency (CNES), Toulouse, France. [10]Center for Remote Sensing Application (CRSA), Mohammed VI Polytechnic University (UM6P), Ben Guerir 43150, Morocco. ✉e-mail: rafael.almar@ird.fr; bouch@hawaii.edu

at global and local scales (i.e., having an effect on the shoreline, such as wave breaking). A global application of existing ocean models coupled with waves-sea level-rivers-sediment (e.g., CROCO[25], Delft3D[26]) is currently out of reach. Thus, physically simplified shoreline models (e.g., CASCADE[27,28], COSMOS[17], ShorelineS[29], LX-Shore[30,31]) have a bright future as potential tools for investigating drivers of shoreline change at regional to global scales.

Coastal and shoreline management increasingly needs to take into account morphological changes that occur on interannual timescales (i.e. few years to a few decades), especially those related to climate variability[32]. It is therefore of paramount importance to determine the dominant factor influencing these changes at these scales[33,34]. At interannual timescales, the focus of global coastal studies has historically been on assessing the response of the shoreline to regional sea-level changes[35,36]. However, while wind-generated ocean surface waves are known to dominate the impact on beaches at short event to sub-annual scales (e.g[37,38].), recent studies have highlighted the contribution of waves to longer-term interannual coastal water levels[5,39,40] and erosion (e.g[21,41,42].). In addition, the often neglected rainfall/river discharge variability has been shown to play a major role on shoreline evolution through sediment[43–45] but also changes in freshwater river discharge have also been reported to influence coastal sea level[46–49].

The massive re-organizations of the atmospheric and oceanic circulation induced by the El Niño Southern Oscillation (ENSO), arguably the most prominent interannual global climate fluctuation[50], has long been known to cause major shifts in weather patterns and therefore produce interannual variations in global sea-level, waves, rainfall and continental freshwater flux to the ocean[51–54], even far from its dominant region of influence[55,56]. While ENSO is known to be unambiguously dominant in the Pacific with strong impacts on the shoreline and coastal ecosystems[41,42,51,57–64], the possible linkages between ENSO and the key drivers of shoreline change at global scale have not yet been fully explored. In particular, the recent rejuvenation of ENSO research has led to many theoretical breakthroughs in understanding its complex and diverse regimes[65]. On a global basis, other climate modes can also significantly modulate coastal drivers in other ocean basins. Despite its strong dependence on ENSO[66], the Indian Ocean Dipole (IOD) can strongly influence climate variability in the Indian Ocean, especially in its western part, which is less influenced by the Pacific Ocean[67], but also beyond the Indian basin[68]. The Southern Annular Mode (SAM, e.g.[69–71],) plays a major role in the climate of the high and mid-latitudes of the Southern Hemisphere. Its signature on ocean surface waves also extends beyond local wind-generated forcing in the Southern Ocean to distant forcing of wave activity and induced changes in coastal sea level in all tropical basins and, to a lesser extent, even in the Northern Hemisphere[72–75]. In the North Atlantic, climate variability is influenced by the North Atlantic Oscillation (NAO), a large-scale atmospheric circulation pattern produced by the difference between the Icelandic Low and Azores High pressure systems strength[76,77]. Overall, the NAO is known to predominantly control the interannual variability of coastal drivers in the North Atlantic at seasonal to interannual time scales[76,78–81] but can, similarly to the SAM, also affect tropical regions in this basin through the propagation of swell remotely generated at high latitudes[75]. While other regional modes of climate variability may also influence shoreline changes, we have deliberately limited our focus here to the most dominant basin-wide climate modes, the two dominant modes of extratropical variability in each hemisphere, SAM and NAO, as well as the dominant modes of tropical interannual climate variability in the Indian Ocean, IOD, and in the Pacific, ENSO, considered in all its spatial diversity and temporal complexity to account for the effect of ENSO seasonal pacing on the interannual pantropical climate variability[82,83]. However, it is still unclear to what extent the combined and individual variability of these climate modes can explain the overall year-to-year

evolution of global coastal drivers and their subsequent effect on shoreline variability.

Here, we aim to address this knowledge gap by combining a new global dataset of satellite-derived monthly shorelines spanning nearly three decades (1993 to 2019) with global data sets of historical coastal sea-level, waves and fluvial inputs (see the conceptual diagram depicting the adopted methodological approach in Fig. S1). Through this analysis, we gain unprecedented insights on the relative contributions of climate-driven variations in these three forcing to observed interannual shoreline change globally, and on how these contributions vary regionally.

## Results and discussion
### Drivers of shoreline change

The complex phenomenon of shoreline evolution results from the combined influence of several oceanic and terrestrial hydro-sedimentary factors, acting and interplaying at various temporal and spatial scales. Here we consider three main drivers of shoreline change: (i) regional sea level, (ii) ocean waves, and (iii) fluvial inputs. We use the waterline as a proxy for shoreline. Since our focus here is on the interannual variability of climate-driven variability at the global scale, we consider the variability of the monthly drivers smoothed with an 8-month window running mean (to remove sub-annual dynamics from our analysis) and through simplified expressions of their dominant contributors. The influence of regional sea level changes on the water line is straightforward: any change in sea level is a change in the mobility of the water line. The regional sea level anomalies (SLA) considered here incorporate contributions from sterodynamic and manometric sea level changes (due to land ice mass loss and terrestrial water storage changes), as well as atmospheric surge—also known as storm surge, which integrates the influence of both wind setup and surface atmospheric pressure effects (corresponding to the so-called Dynamical Atmospheric Correction). Ocean surface waves affect the waterline in two ways, through morphological changes and the sediment budget (erosion/accretion—widely documented by the coastal scientific community, with reference papers such as Yates et al.[84] but also Splinter et al.[85], among many others) but also through their contribution to the coastal water level via the runup (or setup for the time-averaged component;[86] see Melet et al.[39] for a global assessment on interannual timescales). Here, waves are parameterized as the incoming deep-water wave energy flux (cf. Data and Methods). Similarly, we consider the river flows as a proxy representing the continental influence of fluvial inputs. River discharge also has a dual influence on the waterline. The first effect of rivers on a global scale is sedimentary, through the input of solid sediment[45], which strongly determines the sediment budget of coastal cells: decreasing or increasing, for example, is linked to climate-induced variability in precipitation and is responsible for shoreline retreat or advance[44]. Rivers and their changes in freshwater river flow are also known to affect the waterline through changes in coastal water levels (see review[49]) by affecting the density content of the water column (process and observations[49]). It should be noted that our interest here is not in the precise magnitude of the influence of these drivers, which may be influenced by local and complex nonlinear processes (e.g., complex wave transformation on continental shelves[87] and induced coastal morphodynamics and setup), but only in the expression of their interannual variability. It is also likely that such local effects are damped at the spatio-temporal scales considered here. With these assumptions, the 8-months detrended monthly shoreline interannual anomaly (cleared from the monthly mean climatology) $S$ is then formulated as

$$S(x,t) = \alpha(x)\text{Sea Level}(x,t) + \beta(x)\text{Wave Energy}(x,t) + \gamma(x)\text{Riverflow}(x,t) \quad (1)$$

where $x$ and $t$ represent the along-shore and temporal dimensions, respectively, with 0.5° alongshore and monthly resolution,

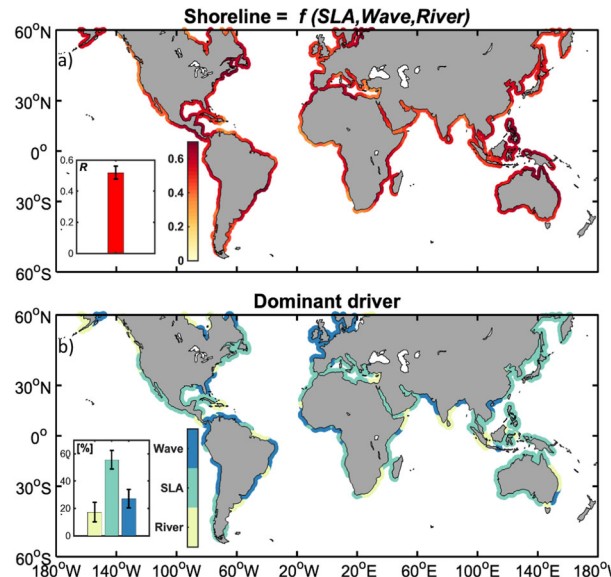

**Fig. 1 | Shoreline change as a linear function of hydrodynamic drivers. a** Global distribution of correlations between interannual anomalies of observed (from Landsat satellite) shoreline position and the multi-linear regression model for shoreline change anomaly (S) as a function of SLA, wave energy flux and river discharge anomalies over the period 1993-2019; only portions of shoreline where correlations are above the 95% confidence threshold are shown. The inset in the bottom left corner shows the globally averaged correlation coefficient. **b** Global distribution of the dominant drivers of modeled Shoreline; a dominant contribution is taken as when > 40% of variance of Shoreline is explained by the variance of one individual driver. The inset in the bottom left corner shows the globally averaged contribution of each driver. For more robustness, whiskers in each inset delineate the range of one standard deviation among all randomized hindcasts of varying lengths from 10 to 27 years.

respectively. For more robustness, the coefficients are calculated based on randomized hindcasts of varying lengths from 10 to 27 years.

Figure 1a shows local correlations between interannual anomalies of shoreline positions S satellite-derived and computed from the multi-linear regression model (Eq. (1)). This comparison yields relatively good model/data correlations that are statistically significant along 91% of the shorelines derived by satellites and 52% of the global world's shorelines (within 60°N–60°S), with a globally averaged correlation of 0.49 (significant at 95% confidence level). Figure 1b shows where each driver dominates the interannual shoreline variability (dominance is

notable exception is along the western North American coastline, where SLA fluctuations associated with zonal oscillations of ENSO over the Pacific Basin dominate interannual variability of the shoreline. Conversely, in enclosed seas such as the Gulf of Mexico, Arabian Sea, the Bay of Bengal or East Sea, average wave action is weaker and the influence of SLA predominates. Unsurprisingly, the influence of fluvial inputs emerge as the dominant driver of shoreline variability near large river mouths (e.g., the Amazon, Niger, Zambezi, Indus and Red Rivers). This is particularly true in the intertropical zone, where, for example, river basins in South Asia (e.g. the Bay of Bengal), South East Asia or North East Australia generally experience strong monsoon-related interannual rainfall variability with significant impacts on continental river flows[88,89]. Another notable exception is the maritime continent (islands, peninsulas and shallow seas of Southeast Asia), where again ENSO-related SLA variability and waves from the strong Northwest Pacific Tropical Cyclone activity[90] appear to dominate shoreline evolution. In the following, we focus on establishing linkages between dominant modes of tropical and extratropical climate variability and the three main drivers of shoreline change considered in our regression model (sea level change, waves, fluvial inputs).

## The influence of ENSO on shoreline driver's climate variability

Assessment of the ENSO teleconnection pathways to these drivers is inherently complicated by the spatial diversity of ENSO in particular related to its two dominant modes of expression, namely the Eastern and Central Pacific El Niño flavors (EP and CP[91,92]), as well as its irregular temporal behavior. The different environmental drivers of shoreline evolution can all be seen as fast transients of the climate system that are constrained by the seasonal and ENSO variability (cf. Data and Methods and Fig S2). Therefore, it is possible to extend the mathematical ENSO-based model of Pacific coastal wave evolution[93] and apply the analytical solution therein to represent the evolution of SLA, wave energy flux and fluvial input globally (Eq. 2). Following the mathematical derivation of the 2nd order solution[94], these relationships can be expressed as the independent multi-linear combination of two indices of ENSO activity, i.e. $E_{mode}$ and $C_{mode}$ that represent the EP and CP El Niño variability, respectively[95], as well as their non-linear interaction with the seasonal cycle (represented by a cosine function with a 12 months period and a phase $\phi$ with a boreal winter peak in January[96]), i.e. the ENSO-annual combination modes $E_{comb\text{-}mode}$ and $C_{comb\text{-}mode}$, known to generate a deterministic variability at near annual time scales as

In order to evaluate the contribution of these distinct ENSO regimes within a more holistic view of global climate variability, we extend this model for each driver to also account for the influence of

$$
\begin{cases}
\text{Sea level}(x,t) = f_1(\text{ENSO}) = a_1(x)E_{mode} + a_2(x)C_{mode} + \left(a_3(x)E_{mode} + a_4(x)C_{mode}\right) * \cos\left(\frac{2\pi(t-\phi)}{12}\right) \\
\text{Wave energy}(x,t) = f_2(\text{ENSO}) = b_1(x)E_{mode} + b_2(x)C_{mode} + \left(b_3(x)E_{mode} + b_4(x)C_{mode}\right) * \cos\left(\frac{2\pi(t-\phi)}{12}\right) \\
\text{River flow}(x,t) = f_3(\text{ENSO}) = c_1(x)E_{mode} + c_2(x)C_{mode} + \left(c_3(x)E_{mode} + c_4(x)C_{mode}\right) * \cos\left(\frac{2\pi(t-\phi)}{12}\right)
\end{cases}
\tag{2}
$$

assumed when the contribution of a given driver is >40%), calculated with respect to the total variance explained. The individual contributions are calculated separately and reduced to the total variance explained by our model, which allows us to accommodate variables that may be partially dependent. While about 50% of the global shoreline (within 60°N-60°S) exhibits a clear dominance of one of the forcing, SLA emerges as the dominant driver of shoreline evolution along most of the global shorelines. However, significant contributions from wave activity is observed along the open west-facing shores such as Western Africa, Western Europe, and Western South America. A

the dominant modes of extratropical climate variability, namely SAM, IOD and NAO, as

$$
\begin{cases}
\text{Sea level}(x,t) = f_1(\text{ENSO}) + \varphi_1(x)\text{NAO} + \delta_1(x)\text{SAM} + \rho_1(x)\text{IOD} \\
\text{Wave energy}(x,t) = f_2(\text{ENSO}) + \varphi_2(x)\text{NAO} + \delta_2(x)\text{SAM} + \rho_2(x)\text{IOD} \\
\text{River flow}(x,t) = f_3(\text{ENSO}) + \varphi_3(x)\text{NAO} + \delta_3(x)\text{SAM} + \rho_3(x)\text{IOD}
\end{cases}
\tag{3}
$$

The distributions of correlation coefficients between observed and simulated (using Eq. 3) interannual anomalies of sea level, wave

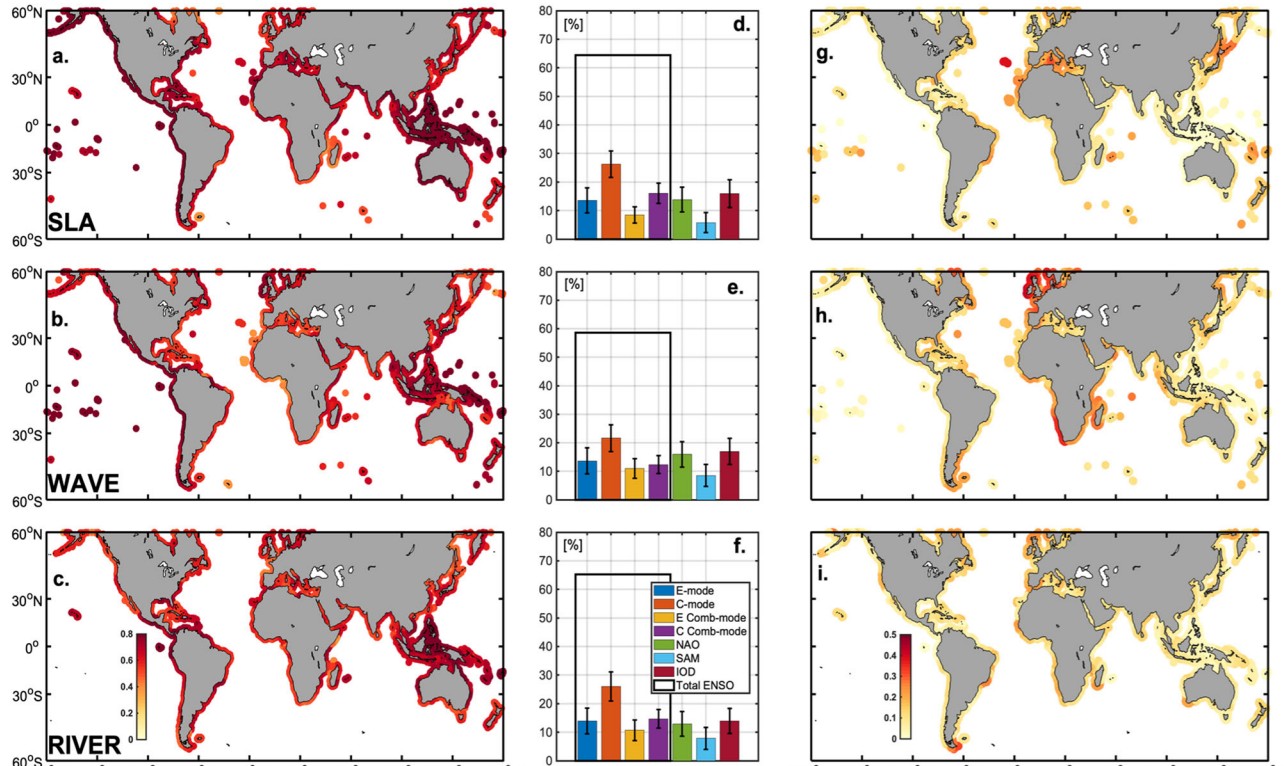

**Fig. 2 | Climate influence on drivers of shoreline change.** Global distribution of correlation coefficients between observed and climate modes-based simulated (Eq. 3) interannual anomalies of sea level (**a.**), wave energy (**b.**) and river discharges (**c.**). Respective percentage of global contributions of the different linear ($E_{mode}$ and $C_{mode}$), non-linear (i.e. combination modes, E Comb-mode and C Comb-mode) ENSO terms, NAO, IOD and SAM to the total model solution for sea level (**d.**), wave energy (**e.**) and river discharges (**f.**). Gain in correlation between observed and simulated interannual anomalies of sea level (**g.**), wave energy flux (**h.**) and river flows (**i.**) respectively associated with the inclusion of NAO, IOD, and SAM into the set of Eqs. (2). Whiskers in each inset delineate the range of one standard deviation among all randomized hindcasts of varying lengths from 10 to 27 years. In panels (**a**), (**b**) and (**c**) only portions of shoreline where correlations are above the 95% confidence threshold are shown.

energy flux and river flows are shown in Fig. 2a, b and c, respectively. The model that accounts for the effect of the four climate modes (ENSO, IOD, NAO, and SAM) produces an overall correlation of 0.72, 0.64, and 0.61 with the reanalysis products of SLA, wave energy, and river flows, respectively (cf. Table 1 of supplementary material), and exhibits statistically significant correlation at the 95% level along the world's shorelines. Figure 2d–f break down the respective contribution of NAO, IOD, SAM, and ENSO to the model's globally averaged total variance. With an average contribution of ~65%, ENSO, in all its diversity and complexity (i.e. Total ENSO in black contour bar), emerges as the predominant driver of global climate-driven interannual variability of the four climate modes considered. This theoretical framework also allows disentangling the respective contributions to the drivers of shoreline change from (i) the two types of ENSO linear forcing, the $E_{mode}$ and $C_{mode}$ and (ii) their non-linear interactions with the seasonal cycle (the combination modes, *E Comb-mode* and *C Comb-mode*, i.e. the last two terms of Eqs. (2)[97]). Over the study period, several El Niño events were recorded: a major CP (2009/2010), and several small CP (2002/03, 2005/06). The linear ENSO effect appears to dominate the interannual anomalies of all shoreline drivers (orange bars in Fig. 2d–f). Because the $E_{mode}$ is mostly related to the extreme El Niño occurring in the far eastern Pacific (e.g., the 1997/98, 2015/16 El Niño), the $C_{mode}$ (prevailing during the study period) overshadows the $E_{mode}$ contribution. Nevertheless, ENSO's non-linear influence is far from negligible, with the *E Comb-mode* and *C Comb-mode* contributing together ~25% to the total variance (cf. yellow and purple bars on Fig. 2d–f).

The contributions from the extratropical climate patterns to the drivers of shoreline change associated with NAO (green bars in

Fig. 2d–f) and SAM (light blue bars in Fig. 2d–f) reach on average 15 and 8% globally, respectively. The IOD contribution (burgundy bars in Fig. 2d–f) is also around 15%. Figure 2g–i shows the gain in correlation between observed and simulated interannual anomalies of sea level, wave energy flux and river flows, respectively, associated with the inclusion of NAO, IOD, and SAM into the set of Eqs. (2). The influence of NAO on all three drivers is strongest in the northern Atlantic and Mediterranean basins. This is due to its strong effect on thermosteric variations[79,98,99], as well as the atmospheric pressure field and meridional gradient anomalies that force the zonal wind field and lead to dynamical sea level and precipitation changes[100,101] as well as increased wave activity in the North and tropical Atlantic[102]. A substantial increase in correlation can also be observed in the Southern Hemisphere (e.g., Indonesia, South Africa, South America), particular in wave energy (Fig. 2h), owing to the influence of SAM on the interannual variability of ocean wave activity south of 30ºS, whereas we can hypothesize that the increase in wave energy correlation along the Eastern African façade is due to effects from the IOD.

Overall, our analysis reveals that accounting for the full continuum of ENSO effects on drivers of shoreline change explains most of their variability with a substantial gain compared to when only its canonical influence is considered as commonly done (measured by a simple linear regression of the shoreline drivers onto the classic Niño3 index, i.e., the usual benchmark in ENSO studies, see Table S1). Our new framework indeed allows considering the wide spatial diversity and temporal irregularity of ENSO teleconnections operating towards higher latitudes (Figs. S3b, e, h) and other oceanic basins (Figs S3c, f, i) whereas the canonical variability tends to limit such atmospheric and

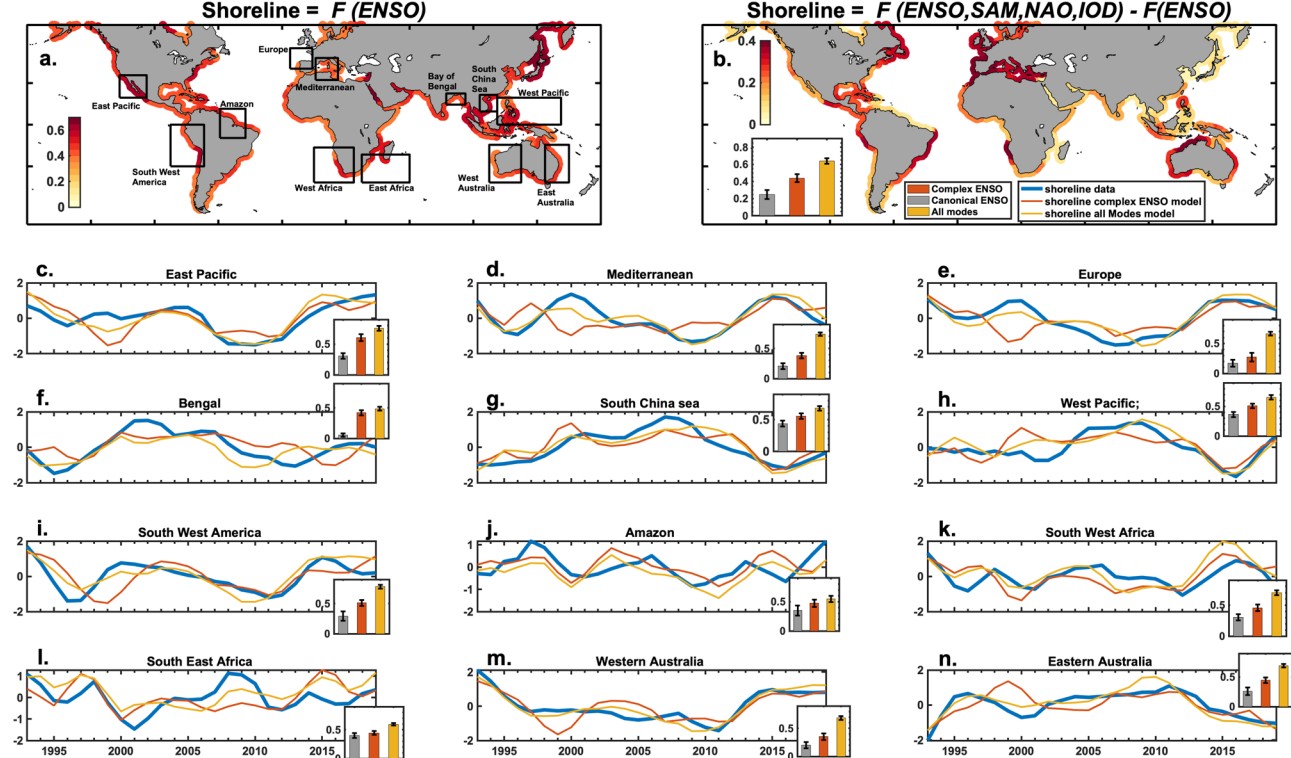

**Fig. 3 | ENSO-based model of interannual normalized shoreline change. a** Global distribution of correlation coefficients between observed (from Landsat satellite) and ENSO-based (Eq. 4) simulated interannual anomalies of shoreline change. Only portions of shoreline where correlations are above the 95% confidence threshold are shown. **b** Gain in correlation between observed and simulated interannual shoreline anomalies associated with the inclusion of NAO, IOD and SAM into Eq. (4). Panels (**c**) to (**n**): Observed and simulated time series of yearly averaged interannual shoreline monthly anomalies averaged over the corresponding regions delineated by the black boxes on the left map when the model considers all climate modes or the complex ENSO only. Inserted bar plots in each time series plots indicate the shoreline change variance explained (in %) by the complex ENSO model (orange bars), by the simple linear regression model onto the canonical ENSO mode (represented by the classic Niño3 index, gray bars) and by the model considering all climate modes (yellow bars). Whiskers in each inset delineate the range of one standard.

oceanic bridges essentially to the tropical Pacific. It is also noteworthy that the inclusion of SAM, IOD and NAO as predictors leads to a similar improvement in the variance explained globally, i.e. +12% for SLA, +16% for wave energy and +11% for river flow for a total variance explained of 52%, 41 and 37%, respectively, with the improvements mainly restricted in term of spatial influence to the shorelines located in the high latitudes of the North Atlantic and Southern Hemisphere with the exception of the eastern tropical African coasts (Table S1).

## ENSO-based model to compute shoreline evolution

The dominant El Niño influence on the key drivers of shoreline evolution over the last three decades highlighted above suggests that ENSO has a very substantial influence on shoreline variability at interannual timescales along most of the world's shoreline. From Eq. (1) and Eq. (2), our mathematical expression can thus be applied directly to compute shoreline change anomalies using only ENSO characteristics as input (Eq. 4):

$$S(x,t) = F(\text{ENSO}) = A(x)E_{\text{mode}} + B(x)C_{\text{mode}} + (C(x)E_{\text{mode}} + D(x)C_{\text{mode}}) * \cos\left(\frac{2\pi(t-\phi)}{12}\right) \tag{4}$$

This simplified ENSO-based model, which integrates the essential environmental factors impacting shoreline variability, simulates the global shoreline anomalies (Fig. 3) relatively well, with a globally averaged correlation of 0.43 (significant at the 95% level). This demonstrates that the ENSO state, when represented in all its spatial diversity and temporal complexity (as compared to only its canonical expression as commonly done) may be a reasonable predictor of the main processes affecting shoreline changes even outside the Pacific basin and the tropics (see the regional and global gain from the canonical to complex ENSO model formulation on the different inset bar plots of Fig. 3 and Table S1). However, while ENSO's influence is significant (at 95% significance or more) along about 83% of the world's shoreline estimated by satellite (47% of the world's total shoreline), there remain several stretches where correlations are below the 95% significance threshold. The inclusion of SAM, IOD and NAO into Eq. (4) results as a new Eq. (5):

$$S(x,t) = F(\text{ENSO}) + \varepsilon(x)\text{NAO} + \zeta(x)\text{SAM} + \eta(x)\text{IOD} \tag{5}$$

This leads to a notable increase in global average correlation, which then reaches 0.62, and in particular along most European and Southern Hemisphere's shorelines. This can be explained by the consideration of atmospheric factors associated with NAO[78,98,103,104] and SAM and their induced effects on SLA and waves on extratropical shorelines in both hemispheres. The increase in variance seen in the Indian Basin is likely related to the inclusion of the IOD and to some extent the SAM, which can generate waves strong enough to travel to these regions (e.g., Indonesia, northwest Australia, India).

NAO and SAM, whose expressions are predominantly atmospheric, offer little seasonal predictability other than that linked to ENSO itself[105]. Similarly, the IOD variability unrelated to ENSO is mostly stochastic[66] and therefore of little value for improving seasonal climate predictions. On the other hand, ENSO, as a slower mode of variability

characterized by a strong subsurface oceanic signature, remains the most reliable predictor for climate forecasts at time scales beyond the seasonal horizon[106]. Our results show that reasonably skillful predictions can be obtained from computationally cheap statistical models approximating the evolution of the diverse ENSO regimes by linear and nonlinear dynamics. We believe the results and methods presented in this study provide a solid conceptual framework to evaluate complex connections between large-scale climate variability and regional coastal hazards and could form the basis for developing regional physics-based projections of shoreline change to feed into effective coastal adaptation strategies around the world.

## Limitations

Several simplifications were necessary to perform this study at global scale, as is the case with any large-scale analysis. We have excluded very high-latitude coasts where the seasonality of environmental variables is perturbed by ice[107]. We have used the global, and necessarily coarse, publicly available ERA5 wave reanalysis product that has nevertheless been widely used and validated in the literature. Additionally, we did not consider the transformation of wave energy from offshore deep waters to the continental shelves and farther into coastal zones, a process that can be substantial in areas with wide shelves[37,87]. Instead, we focused directly on offshore waves and considered only regional patterns of variability under the influence of the synoptic climate environment. Small-scale (spatio-temporal) coastal dynamical processes are complex and out of reach of the large scales covered by this interannual regional to global study. Their physical influence on shorelines is also multifaceted. For instance, the influence of rivers on coastal sea levels[46,47] and sediment discharges[45,108,109] depends on several socio-environmental aspects and such information is still unavailable worldwide despite an urgent need. Our approach is focused on the interannual variability of the influence of shoreline drivers, and not on the precise magnitude of these influences, which can be substantially locally modulated by several processes operating at shorter time scales. We believe the year-to-year assessment in this study dampens such small-scale, non-linear and complex interactions and helps the distillation of large scale, statistical links between shoreline drivers and climate signals such as ENSO. Overall, the correlations obtained between our proxies and the observed/modeled shoreline changes provide reasonable confidence in our approach (Fig. 1).

Here we examine the influence of interannual climate variability on the world's coastlines, excluding longer-term trends from our analysis. This long-term, decadal to centennial, shoreline evolution is influenced by a variety of factors, including trends in our considered drivers (waves, sea level rise and river discharge), but also other factors such as vertical land motion, also called subsidence, due to natural (such as Glacial Isostatic Adjustment (GIA) - or post-glacial rebound) and local human influence. High rates of relative sea-level rise due to subsidence in urban areas such as Jakarta, have been reported to reach tens of centimeters per year, and most deltas are sinking due to oil exploitation and agriculture[110], by far exceeding natural variability and even all projected worst-case scenarios of mean sea level rise over the 21st century[111,112]. Changes in the coastal sediment budget due to terrestrial inputs, climate change and human infrastructure (river damming, coastal protection and deep-water harbors) also play a dominant role in the long ter. On a long-term basis, current wave trends are not expected to continue in the future unlike sea level[10], which is predicted to rise steadily or even accelerate.

Furthermore, our methodology considers natural stretches of coasts influenced by natural climate variability. However, shorelines have been actually modified in various ways by human activities, particularly in urbanized areas where, for example harbors have been constructed, land reclaimed from the ocean[20], seawalls built to combat shoreline recession, cliffs stabilized, beaches nourished, and groins placed in an attempt to retain a beach fringe and maintain dunes. For example, in the US alone, 14% of national shoreline is estimated to be hardened with engineering structures (e.g. seawalls, dikes[113]), and this percentage is expected to intensify globally over the 21 s century[114,115]. Human intervention is particularly high in tropical developing countries, where dramatic changes in land use are occurring, due to, for instance, deforestation and urbanization, at a higher rate than anywhere else in the world[116,117]. In particular, unplanned or poorly designed coastal structures are a major issue transforming the coastal landscape in these countries. The regional aggregation of our data at synoptical scale (8 transects, ~400 km) aims at damping these local and human-induced influences to enhance and capture larger scales climate-driven patterns.

Satellite-derived shorelines can be prone to many sources of uncertainties or systematic biases that can confound analyses such as those presented here. Therefore, it remains challenging to assess whether the absence of a relationship with potential drivers (e.g., ENSO) is due to a true lack of relationship, or simply due to poor quality shoreline data. The monthly median NDWI shoreline mapping approach used here is likely to be more susceptible to potential data issues than other approaches that use longer annual composites[20,118], particularly in regions of the world with either high persistent cloud cover, or relatively low satellite observation densities[119,120]. This can make it challenging to obtain even clean annual median shorelines in many of these low data environments, let alone high-quality monthly shorelines. Similarly to a former global study[20] and unlike Vos et al., (2019)[19] who used advanced trained convolutional neural network coefficients to distinguish between land and marine pixels, here we used the more basic NDWI waterline proxy for this global application. Our shorelines are smoothed in the same way as the drivers over an 8-month period to eliminate sub-annual shoreline dynamics, which also smooths out some of the problems associated with getting good monthly shorelines. The correlations obtained between our independent drivers/climate modes and the observed/modeled shoreline changes provide a reasonable level of confidence in our satellite-based global shoreline dataset that admittedly can be further improved, but paves the way for more future detailed studies and technological developments.

## Methods

Our methodological approach is summarized in the Supplementary material Fig. S1 and detailed in the following.

### Shorelines from satellite images

Here we use the water line as shoreline definition[121], i.e., the water line at the time of data collection. Due to the continuous influence of tides, storm surges and waves on the shoreline, the water line is subject to a combination of sediment and hydrodynamic variabilities that do not directly represent the evolution of the "geological" shoreline, such as the retreat of mean high-water line, the vegetation line, the erosion of a cliff, or the erosion of a coastal settlement. Different portions of the shoreface profile are likely to have contrasting responses to drivers of change, even potentially exhibiting contrasting trends through time and space[122,123]. Nevertheless, the water line adequately reflects the shoreline position that is relevant for vulnerability and risk associated with erosion and flooding[118] and is thus used a shoreline proxy in this study.

The global dataset used in this study is re-sampled with transects spaced at 0.5° intervals (~50 km), along the same 14,410 points vector as in Almar et al.[40] spanning approximately 1.5 million kilometers. The initial shoreline dataset used is Global Self-consistent Hierarchical High-resolution Geography (GSHHG version 2.3.6[124]) to define locations along the world shorelines. The world was divided into computational regions using a series of GSHHG shoreline polygons. This

positions our study not at the local scale (<50 km, ~ specific bay, beach, community seafront), but to capture the regional to global picture. The local coastline has its own complexities (e.g., wave transformation on unknown changing bathymetry, impact of infrastructure and human intervention), which are beyond the scope of this study. Instead, the individual data are more regionally aggregated (along 8 consecutive data points, ~400 km of coastline), showing similar regional behavior rather than distinguishing local diversity (see Fig. S4). The monthly composites of shoreline positions were derived from 1993 to 2019 using multiple satellite acquisitions provided by the Landsat missions 5, 7, and 8. The extraction of these data was performed on the Google Earth Engine (GEE) platform[125]. The GEE was calculated over 30 regions of interest of varying size, covering coastal areas worldwide (and 60% of the globe). Since we used T1_8DAY_NDWI 30 m collections from Landsat 5, 7 and 8 satellites, which represent monthly median composites of 10.5 images (i.e., 3 or 4 images per month depending on the month × 3 satellites) of size 0.70 × global area at a 30 m resolution, about 400 Megapixels were processed, which amounts to approximately three petaoctets of satellite data and required 7200 h of computation. Normal Difference Water Index (NDWI) maps were derived from satellite images and the NDWI threshold used was 0[126]. The identified pixels correspond to ocean for NDWI > 0, and to land surfaces for NDWI < 0. The shoreline is then identified as the interface between the land and sea surfaces[127]. We acknowledge that the selection of water index thresholds can have a significant impact on the quality and distribution of satellite-derived shorelines. The constant NDWI threshold used here contrasts with the use of more complex dynamic methods to optimize thresholds to local conditions (e.g., the commonly used Otsu method[128]), but remains the most commonly used approach to obtain a primary estimate and gives reasonable results at the validation sites.

Issues due to wave breaking or water turbidity during extremes are smoothed out using monthly median composites in addition to the post 8-month smoothing to remove event-related and sub-annual dynamic. Also, our study focuses on interannual evolution, which dampens the complexity of this short-term link between drivers and shoreline evolution (with potential lags[129,130]). To illustrate the ability but also the limitations of our method to observe shoreline variability from satellite, Fig. S6 shows a comparison between the closest satellite transects and various ground measurements of some of the longest shoreline datasets around the world: Truc Vert[131] (South West France, Fig. S6a), Torrey Pines[132] (West Coast USA, Fig. S6b), Duck (East coast USA, Fig. S6c, data provided by the U.S. Army Engineer Research & Development Centre, Coast & Hydraulics Laboratory, Field Research Facility) and Narrabeen[133] (East Coast Australia, Fig. S6d). The in-situ data are based on regular monthly topo-bathymetry measurements averaged along the coast (typically one kilometre), and the comparative shoreline proxy used here is the high tide upper beach contour above mean sea level. For all sites, the ground truth data are interpolated to a regular monthly resolution, and comparisons are made for periods where no significant gaps were present in the in-situ data. Despite the coarse resolution of our dataset (transects every 0.5°), our regional comparison with local measurements shows good overall agreement, increasing from short, seasonal, to longer interannual time scales. The local behavior of the beaches such as a nourishment at Torrey Pines cannot be captured and is beyond the scope of our regional to global analysis. Here, we are after the variability of the shoreline for which the correlation is the most appropriate quality proxy. The correlation is used to assess the quality of our dataset compared to in-situ data. It should be emphasized that we aim here to resolve only the regional to global scales of interannual variability of the coastal shoreline, not the amplitude of the subsequent numerous and diverse processes that may include non-linearities and interactions within the coastal system[37,134]. These correlation coefficients between our satellite-derived shorelines representative of the regional scale,

and in-situ local shorelines range from 0.38 to 0.61 at these sites, despite distances of up to tens of kilometers between our closest transects and the sites. The differences may come from the difference in the shoreline approximation used; thus, all sea level variations, such as regional sea level, wave contribution to sea level at the coast (i.e., setup and run-up) but also river discharge have a more direct effect on the position of the waterline than the surveyed shoreline proxy using mean sea level as a reference. Nevertheless, this demonstrates the regional common behavior of shorelines at interannual scales, already identified[42,57,104].

## Sea level, waves and river flow

Sea level was computed at the coastal points situated along the open coasts of the world. Regional sea level anomaly (SLA) was derived at each computational profile from satellite altimetry sea level time series using the SSALTO/DUACS multi-mission data[135]. In addition, Dynamical Atmospheric Corrections (DAC, or storm surge) were taken from a global application of the hourly MOG2D-G non-structured grid model outputs[136], forced by surface winds and atmospheric pressure from the ERA-interim reanalysis[137]. The offshore wave energy flux, proportional to and here directly taken as $H_s^2 x T_p$ where $H_s$ is the significant wave height and $T_p$ the swell peak period[138] was extracted from ERA5[139], developed by the European Centre for Medium-Range Weather Forecasts model (ECMWF), at 0.25° × 0.25° and hourly temporal resolution. The ERA5 reanalysis uses a coupled ocean wind-wave and atmospheric model, which has been extensively validated[137,140,141]. For continental freshwater river flow data, used here as a proxy for annual variability of fluvial inputs, we rely on daily runoffs from the up-to-date *ISBA-CTRIP* (Interactions between Soil, Biosphere and Atmosphere-Total Runoff Integrating Pathways, from the Centre National de Recherches Météorologiques—CNRM) land surface model simulations[142]. ISBA-CTRIP is a "state-of-the-art" hydrological numerical system that simulates continental hydrology and freshwater river flow at the coast globally. It is based on a two-way coupling between the ISBA and CTRIP models, where the ISBA solves the land surface energy and water budgets at any time step, while the CTRIP river routing model simulates natural river discharges up to the ocean from the total runoff computed by the ISBA land surface model. Here, we use the global offline simulation at 0.5° resolution driven at a 3-hourly timescale by the ERA-Interim (ECMWF Reanalysis) reanalysis available over the 1979–2019 period. At each time step, ISBA-CTRIP provides the variations of continental freshwater flux to the ocean from which we use here yearly average for 1993-2019. Note that ISBA-CTRIP does not include anthropogenic effects on water storages and river discharges since it does not include representations of flow regulation and irrigation water needs which can have a profound impact[45,143,144]. Nevertheless, modeled daily runoff has been extensively validated against several database of in-situ daily measurements for large rivers in different environments[142], showing good accuracy and agreement in terms of seasonal and interannual variations.

All these data were interpolated on to the shoreline transect locations using the nearest neighbor method and computations were performed at all 14,140 shoreline transect locations. In order to eliminate local effects and to focus on variability at regional scales, all sea level components were spatially smoothed such that all calculations used median values within a radius of 100 km alongshore. All data were linearly detrended, the seasonal cycle removed using a monthly mean seasonal climatology and to focus on the interannual variability smoothed using a running mean with an 8-month window over the period 1993–2019.

## Toward an ENSO-based shoreline prediction

Here, to construct a model of shoreline change drivers, we follow Boucharel and Jin's[95] approach, based on the stochastically forced

model of fast climate variability[145,146]. We consider the variations of coastal wave energy flux, sea level and river inputs as fast transients of the climate system that respond to slow climate forcing, i.e., the two different types of ENSO and their seasonally modulated influence on tropical and extra-tropical storm activity, precipitation and SLA regional patterns. The seasonal connections between ENSO and these drivers of shoreline evolution are evidenced by the regression patterns between interannual anomalies of precipitation, sea surface temperature (SST), wind speed and direction and the $E_{mode}$ and $C_{mode}$ in boreal winters and summers (Fig. S2). $E_{mode}$ and $C_{mode}$ are two uncorrelated and independent ENSO indices, calculated as the first two rotated Principal Components of the EOF decomposition of SST interannual anomalies[94] and accounting for the variability of the two different types of ENSO, respectively, the extreme warm events in the Eastern (i.e. EP El Niño) and moderate warm events in the Central Pacific (i.e. CP El Niño). Note that the two classical ENSO indices Niño3 (monthly sea surface temperature anomalies averaged in the region bounded by 5°N to 5°S, from 170°W to 120 °W) and Niño4 (monthly sea surface temperature anomalies averaged in the region bounded by 5°N to 5°S, from 150°W to 90°W) are almost identical to the $E_{mode}$ and $C_{mode}$ indices, respectively, and provide a similar but simpler and more direct measure of ENSO spatial diversity, although not quite orthogonal. Nevertheless, we re-ran all the main calculations and figures using these simple indices, which gave similar results (see Figs. S5, S6). Both types of El Niño events are associated with strong zonal swings in SST anomalies across the tropical Indian and Pacific basins, which have massive repercussions in terms of coastal sea level variability in the tropical band through thermosteric effects. This well-known ENSO zonal redistribution of ocean heat can also induce, via atmospheric teleconnections[55], a strengthening of the North Pacific jet-stream that can even extend to the North Atlantic basin at the peak of EP El Niño events (Fig. S2a). This will increase the coastal wave activity along west-facing shorelines via the intensification of the Aleutian (Icelandic) low-pressure systems in the North Pacific (Atlantic) basins. This strengthening of surface winds will also affect coastal sea level along the Northern Hemisphere west-facing shorelines through dynamic effects. Because ENSO-driven changes in SST also affect the Walker circulation, which induce significant redistribution of deep atmospheric convection, ENSO has a strong influence on large-scale precipitation patterns and in particular on the intensity of monsoonal regimes. The onset of El Niño is associated with drier conditions over South Africa, the Maritime continent and the Middle East, which extends over the entire tropical Atlantic during CP events (Fig. S2c, d). At their peak, the East Asian, North American and West African monsoons are significantly strengthened while the South American monsoon is weakened. These changes in rainfall patterns are generally translated to the amount of freshwater discharged by rivers to the ocean. Overall, this analysis confirms that, similarly to wave variability in the Pacific, ENSO has a substantial seasonally-modulated influence on SLA and river discharge variability as well, particularly in the Pacific and the tropics. Therefore, we can use the same hypothesis and expand the model presented by Boucharel et al.[93] and Boucharel and Jin[95] for coastal wave activity also for SLA and river flow. The analytical solution of low-frequency waves, SLA and river discharge amplitude changes ($Z$) can be then written in a general form as

$$Z = 1 + k_1 m(t) + k_2 m^2(t) + ... \qquad (6)$$

with $m(t) = \gamma_A \cos\left(\frac{2\pi(t-\phi)}{T_A}\right) + \gamma_C C_{mode} + \gamma_E E_{mode}$ representing the climate forcing (first term being the seasonal cycle with a period $T_A = 12$ months and a phase similar to that of ENSO peaking in December–February, $\phi = 1$ (i.e. January) and the last two terms the CP and EP El Niño forcing, respectively).

At the 2nd order, the interannual evolution of the amplitude of the driver $Z$ can then be expanded (Eq. 6) as in Eq. (2) or Eq. (4). Since our

model resembles Linear Inverse Models, we can obtain the coefficients of the different terms, and therefore, the full analytical solution through a local multi-linear regression[147,148].

We compare this model that integrates the spatial diversity as well as the time scales associated with the different ENSO-annual cycle combination modes to what is commonly known as the canonical ENSO effect, the current state-art-of-the-art or benchmark in studies of coastal impacts of ENSO; a simple regression model onto the classic Niño3 index.

## Other climate modes

To differentiate between ENSO and other climate modes, we use the Indian Ocean Dipole (IOD), the Southern Annular Mode (SAM) in the Southern hemisphere and the North Atlantic Oscillation (NAO) in the Northern hemisphere. The SAM index is calculated as the zonal pressure difference between mid-latitudes (40°S) and higher latitudes (65°S) of the Southern Hemisphere. The NAO index is measured as the difference in atmospheric pressure at the surface sea level between the subpolar low pressure in Iceland and the subtropical high pressure in the Azores. The IOD is represented by anomalous SST gradient between the western equatorial Indian Ocean (50E-70E and 10S-10N) and the south eastern equatorial Indian Ocean (90°E-110°E and 10°S-0°N). All climate indices were linearly detrended, and the seasonal cycle was removed using a monthly mean seasonal climatology, allowing us to focus on the interannual variability, which was here smoothed using a running mean with a 8-month window over the period 1993-2019.

## Statistical significance of correlations and intervals of confidence

Here, we consider a total period analysis spanning 324 months but with a temporal smoothing using an 8-month window running mean, which leads to $324/8 = 40$ independent time steps. In Fig. S5, we estimated the slope of the best linear fit to the autocorrelation logarithm for each driver and the shoreline, or in other terms, an e-folding time equivalent to their interannual memory. While such memory times vary differently for the drivers and the shoreline position along the different global shorelines, they remain below a maximum of ~24 months. In addition, because our estimation of the shoreline from space is based on the waterline position, we expect an instantaneous response of shoreline to SLA variation. Studies in Australia[149] and in France[129] show with high frequency observations that the memory of shoreline with respect to the wave forcing does not exceed two weeks. Although the time scales of the shoreline's response to the input of sediments from river discharge remains somewhat uncertain, a local lead-lag correlation analysis between the shoreline variability and its dominant driver's shows that the maximum coherency always occurs at a lag below 12 months (Fig. S8). Since we explore the interannual variability of the shoreline, this implies an in-phase relationship between its position and all hydro-dynamic forcing.

This leads to a total number of independent observations for shoreline and drivers of $N = 40 - (24/8) = 37$. Thus, the total number of degrees of freedom d.o.f for a multiple linear regression analysis with k predictors is d.o.f = $N$-$k$-$1$. For instance, this gives a d.o.f = 37-3-1 = 33 and correlation coefficient thresholds of 0.32 and 0.44 at the 95 and 99% significance level, respectively, according to a Student $t$ test for Fig. 1. For Fig. 2, d.o.f = 37-7-1 = 29 and correlation coefficient thresholds are 0.36 and 0.46, respectively.

To provide intervals of statistical confidence, we compute coefficients of the multi-linear regression hindcast model at each coastal point over various sub-periods ranging between 10 and 27 years (depending on available data series lengths) from the total 1993-2019 period. For instance, for a hindcast of a given length of $n$ months, we realize $324 - (n - 1)$ randomized hindcasts.

The variance explained by different contributors (i.e., driver or climate mode components) to the total regression model is calculated as (Eq. 7):

$$\text{explained variance} = 100 * \left(1 - \frac{\text{var(total model} - \text{contributor)}}{\text{var(total model)}}\right) \tag{7}$$

## Data availability

The raw climate data that support the findings of this study are already available online. AVISO (https://www.aviso.altimetry.fr/en/data/products/auxiliary-products/dynamic-atmospheric-correction/description-atmospheric-corrections.html), ERA5 (https://cds.climate.copernicus.eu/cdsapp#!/dataset/reanalysis-era5-single-levels?tab=overview), NOAA climate indices (https://psl.noaa.gov/data/climateindices/list). ISBA-CTRIP (http://www.umr-cnrm.fr/spip.php?article1092).

## Code availability

Matlab codes and processed data are made available upon request.

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

## Acknowledgements

The authors thank the Google Earth Engine team for enabling this study with the tools, computational resources, and satellite imagery (Landsat) provided free of charge. R.A. supported by the French Agence Nationale de la Recherche project "GLOBCOASTS" (ANR-22-ASTR-0013). R.R. is partly supported by the AXA Research Fund. J. Boucharel is funded by the project MOPGA "Trocodyn" (ANR-17-MPGA-0018) and wants to thank the Région Occitanie.

## Author contributions

R.A. and J.B. designed and conceptualized research. R.A., J.B., and M.G. performed research. E.W.J.B. participated in early discussions and analysis. F.P. analyzed river discharge. M.W.B., G.A., G.T., and M.G. implemented and run the GEE scripts. G.A., G.T., J.M., and M.G. post-processed the GEE-extracted data. J.B. contributed new reagents/analytic tools. R.R. and F.F.J. provided strategic advice on fine tuning the study and analysis. All authors discussed the results and contributed to writing the manuscript.

## Competing interests

The authors declare no competing interests.
