## [Peer Review File · Nature Communications]

Influence of El Niño on the variability of global shoreline positionEditorial Note: Parts of this Peer Review File have been redacted as indicated to remove third-party material where no permission to publish could be obtained.

REVIEWER COMMENTS

Reviewer #1 (Remarks to the Author):

Review of El-Nino paper

This paper is interesting and at a global scale it demonstrates a strong association between coastal shoreline change/variability and variation in sea level, wave energy and river discharge. Further this can be linked to climatic variations and especially El-Nino. As this is the first demonstration of this relationship at a global scale, I find this to be an important paper which builds on earlier broad-scale – but sub-global analyses -- such as Vos et al (2021) on the Pacific. However, there are some clarifications and improvements that are needed before it is suitable for publication.

The paper shows an association between the three drivers and shoreline variability. A conceptual model is mentioned but how exactly does this model work – or rather how do the authors assume that the three drivers operate and what conditions promote shoreline retreat and shoreline advance? This should be explained in detail in additional material associated with the supplemental material – around Figure S1. Once this is written the main material should be reviewed and could also be made clearer. I am particularly unclear about the role of river discharge on controlling shoreline position -- the authors should make their conceptual model of shoreline response to higher or lower river discharge much clearer to the reader. Limitations are discussed quite well, but I am especially concerned about how the river discharge driver is assumed to work – is this purely a hydrodynamic effect over the 17 years or is there an influence of the changing sediment budget?.

The paper overly focusses on the term erosion, when it is really talking about shoreline variability, and this includes both erosional and accretional trends – as shown in Figure S4. Similarly, the term coastline and shoreline are used interchangeably in the paper. What is being considered in the paper really concerns the shoreline (line 294) and while linked to the notion of coastline the key term in the paper should be shoreline and change in shoreline position. Beyond addressing these specific points, the authors should make sure their nomenclature and notation is consistent and clear.

The results that are presented here are interesting but qualitatively not surprising based on earlier scholars on coastal evolution who recognised that different drivers would be expected to control coastal evolution at different scales. There are many papers and papers which explore these issues such as the classical textbook by Woodroffe and the more applied work by Stive. More historical acknowledgement of these earlier contributions is needed at least in the Supplemental Material. Some more detailed comments are as follows referenced by line number.

Title is a vague “The influence of El Niño on global shorelines” – suggest “The influence of El Niño on global shoreline position” or something similar that is more precise on what the paper considers.

Abstract should mention how the drivers operate.

19 “coastlines” or shorelines?

19 interannual evolution ‘of shoreline position’ – again need to be more precise

25 “contrasted influence”?? what does this mean? – interacting influence? Improve.

25 “3 main drivers” – change to ‘three main drivers’ – small numbers such as 3 should be stated as a word – three – throughout the manuscript.

34-35 – what are the drivers of increased risk? Partly sea-level rise and climate change, but declining fluvial sediment input as in the Nile delta for example is also a huge effect and widespread?

40 The appropriate SROCC and AR6 chapters would be good additional references here.

47-50 – The focus here is on physics-based models and only one driver?? Climate change – physics-based models should allow consideration of all relevant drivers, including climate change.

303 – leading to how many points on the world’s coast? – 14,410 points (line 307)? Is this a good enough sample of the world’s coast as it is roughly a measurement every 100 km of shoreline length? Discussion of representativeness and scale is needed.

334 to 335 – is the same true for the Sand Engine in the Netherlands? What about the dynamic preservation of the whole coast of the Netherlands where the 1990 shoreline is maintained by law and nourishment is a process like the three factors considered in this paper? A comment is needed.

344 to 348 – sea level? So this is really climate-induced sea level and its variability – this needs to

be stated as such. Many analyses include vertical land movement such as GIA which is excluded here. A clear statement of the implications and limitations is needed here.
352 to 365 – run-off? Can you illustrate the method with an example of the Nile where the run-off to the Mediterranean is now essentially zero due to human use within the catchment. This is not uncommon in arid regions like the Indus or the Colorado in USA/Mexico. Human influence on run-off is profound in some large catchments – how is this represented in the model or is this not considered? This needs to be mentioned and may be a major uncertainty.
641 to 642 – “large” to ‘Large’
794 – why is the classic Duck station used? No worries on this point but it is an excellent dataset.
844 – reference is incomplete

Reviewer #2 (Remarks to the Author):

This is a study of how ENSO dominates year to year changes in coast lines over the period 2000-2017. The authors use a global dataset of satellite derived coast lines, plus measures of sea level variability, surface wave activity and river discharge, to show how the latter three are major controls on interannual coast line variability. They then look at measures of ENSO, SAM, and NAO variability to show that of these three, ENSO is the dominant global control as manifest through sea level, surface waves, and river discharge. This result makes sense, since ENSO is the only phenomenon used as a predictor of coast line variability that has a global fingerprint. The subject is interesting, the paper is well written and organized, and the results are reasonably convincing. However, there are several questions that require clarification before publication.

One is that the authors retain ENSO C-tones but ignore other regional modes of variability. In particular, based on only one reference, they ignore the IOD as largely ENSO driven. While IOD and ENSO are correlated, many other references could be marshaled to argue that dynamics internal to the Indian Ocean also play a role. That was obvious in the record breaking 2019 IOD that was associated with little if any ENSO forcing from the Pacific. Like the other regional modes that the authors do include, e.g. SAM and the NAO, the IOD will have a regional impact in the Indian Ocean. There are other regional modes with year to year variations as well that could be invoked, like the North Pacific Gyre Oscillation, the Atlantic interhemispheric gradient mode, etc. which regionally would account for as much if not more variability than the ENSO C-tones. So it strikes me as unbalanced if C-tones with their very small % variance explained are included and argued to be important, but not any of these other potentially influential climate modes.

The authors argue that including ENSO complexity is important. I agree, but comparisons of a “canonical” ENSO index (NINO3) and E- plus C- indices is cheating to a certain extent, since by now ENSO diversity/complexity is well known territory. A more interesting approach would be the following: NINO3 and the E-index are nearly identical; NINO4 and the C-index are also nearly identical (e.g., Cai et al, 2021, Nature Reviews, Earth and Environment). You can just as easily include ENSO complexity by using the NINO3 and NINO4 indices. In fact, there are advantages of using simple indices like NINO3 and NINO4 which do not require higher order mathematics. How would the results change if these indices were used instead of the E- and C-indices?

I do not profess to be a specialist in shoreline morphology and evolution, so I have what may appear to be a naive question. All the relationships between predictors and predictand (coast line) are presumed to be at zero lag on interannual time scales. However, intuitively, I would expect some of these predictors (e.g., surface waves) to have a cumulative or lagged effect. There should be more discussion of the assumption that everything varies in phase. There appears to be some of that in the section on degrees of freedom, but decorrelation time scale is not quite the same as lead-lag relationships.

Other:

*The figures have terrible resolution and are in some cases impossible to read.

*The authors incorrectly state that the E-index and C-index are uncorrelated. The first and second EOFs of SST are uncorrelated. Their linear combinations in the form of the E- and C-indices are

not.

*In regression analysis, degrees of freedom should be adjusted for the number of predictors. I did not see any discussion of this topic and how it affect the uncertainty of the estimated parameters.

*How accurate are individual estimates of shoreline position from space? The year to year changes as shown in Figure S4 are $\pm 5-10$ m, which is not large, so the noise to signal must be significant.

Reviewer #3 (Remarks to the Author):

Review of: The influence of El Nino on global shorelines

By: Rafael Almar, Julien Boucharel, Marcan Graffin, Gregoire Ondo Abessolo, Gregoire Thoumyre, Fabrice Papa, Roshanka Ranasinghe, Jennifer Montano, Erwin W.J. Bergsma, Mohamed Wassim Baba and Fei-Fei Jin

I have reviewed the above manuscript and find that it will be suitable for publication in Nature Communications following moderate revisions. The manuscript is very well written and therefore I am not submitting an annotated version of the manuscript. Below please find general and specific comments concerning possible suggestions for the authors to address.

General Comments:

The paper "The influence of El Nino on global shorelines", presents a global data set of satellite derived shorelines, a conceptual model of the primary drivers of the shoreline evolution captured by the data, and a quantitative framework exploring the role of ENSO variability in forcing their interannual variability in both the drivers and the morphological response.

My first general comment is regarding the uncertainty of the satellite derived shorelines (SDS). Recent studies have shown that the uncertainty of SDS varies with shoreline type and metocean forcing conditions and therefore must vary widely globally and with time (satellite mission). A more quantitative handling of SDS uncertainty is warranted. For example, the four sites chosen to assess the quality of the SDS data all are areas that are not subjected to persistent cloud cover. Most likely SDS uncertainty in regions that do have significant cloud cover would be much worse. Further, the model developed in this paper attempts to explain the detrended monthly shoreline anomaly – with sub annual dynamics explicitly removed. The 'good agreement' on the SDS quality plots shown in Figure S4 have reasonable correlation precisely because the seasonal signal is left in. It is not clear, at least to me, how well the filtered SDS data (8-months window running mean) would recover the similarly filtered shoreline signal from in situ data.

I am also concerned that perhaps the abstract oversells the paper's contribution. The statement: 'Our results represent a new framework for understanding and predicting climate driven coastal hazards' is odd considering how much work has made to identify a direct connection between ENSO and coastal change over the last several decades. In fact, a general comment is that the paper lacks a thorough set of citations regarding the influence of ENSO on shorelines. At present there are more citations focused on the role of the NAO on shoreline variability than citations listed focused on the already demonstrated role of ENSO.

The remainder of the paper is quite strong, and the approach presented for disentangling the relative influence of the various climate driver flavors is a significant contribution.

Specific Comments on Text:

Line 47-50: This is a personal preference of mine, but why present these statements in a negative light (using words like 'not', and 'prevents'...)? The historical global assessments that you are referring to are a huge, very recent advance and provide the opportunity for the analyses you are presenting with this paper. It seems easy enough to frame your contribution as an evolution in knowledge rather than simply filling in what others did not think to do.

Line 54: Oddly worded. Certainly hundreds (perhaps thousands) of studies have 'highlighted the contributions of wind-generated ocean surface waves...' to 'erosion'.

Line 131: Change evolution to variability??

Line 136: Change 'governs' to 'govern'.

Eq 3: better to not reuse gamma here?

Lines 264-265: What about the possibility that the uncertainties in satellite derived shorelines are actually responsible for the significant relationships?

Line 274: change to 'smooths'

Line 278-285: Not mentioning beach nourishment in this discussion seems like a significant oversight.

Line 314: What do you mean by OTSU? Please elaborate or provide a citation.

Line 317: Change to 'dynamics'

Line 341: not sure what is meant by 'transect closets'??

Line 346: Need a '.' after '...2016)'

Line 353: Provide some citations of this extensive validation.

REVIEWER COMMENTS

Reviewer #1

C1-1 This paper is interesting and at a global scale it demonstrates a strong association between coastal shoreline change/variability and variation in sea level, wave energy and river discharge. Further this can be linked to climatic variations and especially El-Nino. As this is the first demonstration of this relationship at a global scale, I find this to be an important paper which builds on earlier broad-scale – but sub-global analyses -- such as Vos et al (2021) on the Pacific.

R1-1 We thank the Reviewer for his/her encouragement and we appreciate his/her suggestions to cite earlier sub-global analyses such as, and among others, Vos et al., (2021) (preprint at the time of the initial submission, now published in Nature Geoscience). It was actually already cited in the previous version of our manuscript but not adequately mentioned in key sections of our manuscript. These earlier important works are now more visible and better acknowledged in our revised manuscript.

This reference was updated in the manuscript to Vos et al., 2023:

Vos, K., Harley, M. D., Turner, I. L., & Splinter, K. D. (2023). Pacific shoreline erosion and accretion patterns controlled by El Niño/Southern Oscillation. *Nature Geoscience*, 16(2), 140-146.

These previous works are now cited in a new paragraph:

“While ENSO is known to be unambiguously dominant in the Pacific with strong impacts on the shoreline and coastal ecosystems (Storlazzi and Griggs, 1998; Ranasinghe et al., 2004; Biribo and Woodroffe, 2013; Barnard et al., 2011, 2015, 2017; Harley et al., 2017; Young et al., 2018; Cuttler et al., 2020; Duke et al., 2022; Vos et al., 2023), other climate modes can also significantly modulate coastal drivers in other ocean basins. Despite its strong dependence on ENSO (Stuecker et al., 2017), the Indian Ocean Dipole (IOD) can also strongly influence climate variability in the Indian Ocean, especially in its western part, which is less influenced by the Pacific Ocean (Marchant et al., 2007), but beyond the Indian basin (Saji and Yamagata, 2003).”

Newly cited literature:

Duke NC, Mackenzie JR, Canning AD, Hutley LB, Bourke AJ, Kovacs JM, et al. (2022) ENSO-driven extreme oscillations in mean sea level destabilise critical shoreline mangroves—An emerging threat. *PLOS Clim* 1(8): e0000037. <https://doi.org/10.1371/journal.pclm.0000037>

Storlazzi, Curt D., and Gary B. Griggs. “Influence of El Niño-Southern Oscillation (ENSO) Events on the Coastline of Central California.” *Journal of Coastal Research*, 1998, 146–53. <http://www.jstor.org/stable/25736131>.

Vos, K., Harley, M. D., Turner, I. L., & Splinter, K. D. (2023). Pacific shoreline erosion and accretion patterns controlled by El Niño/Southern Oscillation. *Nature Geoscience*, *16*(2), 140-146.

Biribo, N., & Woodroffe, C. D. (2013). Historical area and shoreline change of reef islands around Tarawa Atoll, Kiribati. *Sustainability Science*, *8*, 345-362.

Mortlock, Thomas R., and Ian D. Goodwin. "Impacts of enhanced central Pacific ENSO on wave climate and headland-bay beach morphology." *Continental Shelf Research* 120 (2016): 14-25.

Barnard, P. L. et al. The impact of the 2009–10 El Niño Modoki on US West Coast beaches. *Geophys. Res. Lett.* <https://doi.org/10.1029/2011GL047707> (2011).

Barnard, P. L. et al. Coastal vulnerability across the Pacific dominated by El Niño/Southern Oscillation. *Nat. Geosci.* **8**, 801–807 (2015).

Barnard, P. L. et al. Extreme oceanographic forcing and coastal response due to the 2015–2016 El Niño. *Nat. Commun.* **8**, 14365 (2017).

Ranasinghe, R., McLoughlin, R., Short, A. & Symonds, G. The Southern Oscillation Index, wave climate, and beach rotation. *Mar. Geol.* **204**, 273–287 (2004).

Young, A. P. et al. Southern California coastal response to the 2015–2016 El Niño. *J. Geophys. Res. Earth Surf.* **123**, 3069–3083 (2018).

Cuttler, M. V., Vos, K., Branson, P., Hansen, J. E., O’leary, M., Browne, N. K., & Lowe, R. J. (2020). Interannual response of reef islands to climate-driven variations in water level and wave climate. *Remote Sensing*, *12*(24), 4089.

C1-2 However, there are some clarifications and improvements that are needed before it is suitable for publication. The paper shows an association between the three drivers and shoreline variability. A conceptual model is mentioned but how exactly does this model work – or rather how do the authors assume that the three drivers operate and what conditions promote shoreline retreat and shoreline advance? This should be explained in detail in additional material associated with the supplemental material – around Figure S1. Once this is written the main material should be reviewed and could also be made clearer. I am particularly unclear about the role of river discharge on controlling shoreline position -- the authors should make their conceptual model of shoreline response to higher or lower river discharge much clearer to the reader. Limitations are discussed quite

well, but I am especially concerned about how the river discharge driver is assumed to work – is this purely a hydrodynamic effect over the 17 years or is there an influence of the changing sediment budget?

R1-2 Thanks to the Reviewer's remark, more details on the model were introduced in the manuscript. The mechanisms behind the physical link between these drivers and their impact on shoreline are described, with references, in the Introduction. To make it clearer, and following the suggestion made by the Reviewer, this is also described when introducing our conceptual model. Here, we use the monthly waterline as a proxy for shoreline. Ocean surface waves affect the waterline in two ways, through morphological changes and the sediment budget (erosion/accretion – widely documented by the coastal scientific community, with reference papers such as *Yates et al., 2007*; but also *Splinter et al., 2014*, among many others) and also through their contribution to the coastal water level via wave setup/runup (see *Dodet et al., 2019* for a review). For the purposes of our study, waves are parameterized as the incoming deep-water wave energy flux (cf. Data and Methods). River discharge also has a dual influence on the waterline. The first effect of rivers on a global scale is sedimentary, through the input of solid sediment (e.g., *Syvitski and Milliman, 2007*), which strongly determines the sediment budget of coastal cells: decreasing or increasing of fluvial sediment supply to the coast, for example, is linked to climate-induced variability in precipitation and is responsible for shoreline retreat or advance respectively (e.g., *Bamunawala et al., 2021*). Rivers and their changes in freshwater flow are also known to affect the waterline through changes in coastal water levels (for a review, see *Durand et al., 2019*) by affecting the density content of the water column (process and observations, e.g., *Piecuch et al., 2018*). Lastly, the influence of regional sea level changes on waterline is more straightforward, since any varying sea level implies waterline mobility. Of note here, this study seeks to describe the temporal variability of the waterline, and not the amplitude of change, which depends on several factors such as coastal slopes, grain size, but also how the regional drivers used downscale to the coast. Additionally, our study focuses on natural variability at regional scale, and does not aim to capture and resolve various complex human influences, such as river damming and local infrastructures. This is beyond the scope and the reach of our study as it would require data at much finer resolution (~ sub-kilometric alongshore) and accuracy (requiring to conduct our study on individual satellite images, to correct for swash and tide, which would be extremely computationally expensive to perform worldwide). The regional aggregation of our data at synoptical scale (~400 km) aims at damping these local and human-induced influences to enhance and capture larger scale climate-driven patterns. As described in the Limitations section, the yearly aggregation of our results and the focus on the interannual behavior avoid issues that are difficult to address such as lags between drivers and waterline (e.g. storm memory effect at beaches that can reach weeks – see *Angnuureng et al., 2017*). For this revision, we were able to extend all datasets by 9 years, resulting in a 27 year study period, increasing the previous study period by almost 50%. This increases the statistical significance of our results and reinforces our conclusions.

All these points were incorporated throughout the different sections of the revised manuscript, in the Introduction, the conceptual model description, and the Data and Methods and Limitations Sections.

C1-3 The paper overly focusses on the term erosion, when it is really talking about shoreline variability, and this includes both erosional and accretional trends – as shown in Figure S6. Similarly, the term coastline and shoreline are used interchangeably in the paper. What is being considered in the paper really concerns the shoreline (line 294) and while linked to the notion of coastline the key term in the paper should be shoreline and change in shoreline position. Beyond addressing these specific points, the authors should make sure their nomenclature and notation is consistent and clear.

R1-3 Here the Reviewer refers to 2 important points. First, we agree that the use of the term “shoreline position” is more accurate than the term “coastline” in the context of our study and we have now fixed the occurrences of coastline to shoreline accordingly throughout the manuscript. In fact, as mentioned in our previous response, here we use the waterline observed from satellite as a proxy for the shoreline, which has numerous definitions, depending on the context (see Boak and Turner, 2005). The term shoreline can also refer to the mean sea level position (or, among others; high or low tide watermark), which in our case does not make sense as we account for interannual sea level variations. Here, we defined our proxy for the shoreline upfront so that there is no confusion due to differences in definitions of the shoreline. In our study, the waterline given by the satellite water index (NDWI) is the dry-wet limit and is affected by both sea level changes and sedimentary aspects. It is more directly influenced by changes in sea level than the mean sea level contour. More fundamentally, this raises questions on what is the best proxy for shoreline change over the long term, including sea level rise: and using the waterline is advantageous (see the interesting debate by Vousdoukas et al., 2020/Cooper et al., 2020 in Nature Climate Change).

References:

Boak, E. H., & Turner, I. L. (2005). Shoreline definition and detection: a review. Journal of coastal research, 21(4), 688-703.

Cooper, J. A. G., Masselink, G., Coco, G., Short, A. D., Castelle, B., Rogers, K., ... & Jackson, D. W. T. (2020). Sandy beaches can survive sea-level rise. Nature Climate Change, 10(11), 993-995.

Vousdoukas, M. I. et al. Sandy coastlines under threat of erosion. Nat. Clim. Chang. 10 260–263, DOI: <https://doi.org/10.1038/s41558-020-0697-0> (2020).

C1-4 The results that are presented here are interesting but qualitatively not surprising based on earlier scholars on coastal evolution who recognised that different drivers would be expected to control coastal evolution at different scales. There are many papers and papers which explore these issues such as the classical textbook by Woodroffe and the more applied work by Stive. More historical acknowledgement of these earlier contributions is needed at least in the Supplemental Material.

R1-4 We thank the Reviewer for this comment. A paragraph giving deeper background on earlier works was added in the revised version including several key references including Stive et al., 2002, Woodroffe 2002, van

Maanen et al., (2016) and others. We hope that these works (and more) are better acknowledged in the revised version.

Added references:

Stive, M. J., Aarninkhof, S. G., Hamm, L., Hanson, H., Larson, M., Wijnberg, K. M., ... & Capobianco, M. (2002). Variability of shore and shoreline evolution. *Coastal engineering*, 47(2), 211-235.

Van Maanen, B., Nicholls, R. J., French, J. R., Barkwith, A., Bonaldo, D., Burningham, H., ... & Walkden, M. J. (2016). Simulating mesoscale coastal evolution for decadal coastal management: A new framework integrating multiple, complementary modelling approaches. *Geomorphology*, 256, 68-80.

Woodroffe, C. D. (2002). *Coasts: form, process and evolution*. Cambridge University Press.

Some more detailed comments are as follows referenced by line number.

C1-5 Title is a vague “The influence of El Niño on global shorelines” – suggest “The influence of El Niño on global shoreline position” or something similar that is more precise on what the paper considers.

R1-5 We have now amended the title as suggested. Thank you.

The new title reads:

“Influence of El Niño on the variability of global shoreline position”

C1-6 Abstract should mention how the drivers operate.

R1-6 We followed this good advice and added the below sentence in the abstract:

“While sea level has a direct impact on shoreline mobility, waves affect both erosion/accretion and total water levels, while rivers affect coastal sediment budgets and salinity-induced water levels”.

C1-7 19 “coastlines” or shorelines?

R1-7 We thank the Reviewer and as mentioned in a former response we decided to rename all the occurrences of “coastlines” by “shorelines”, which we have also now clearly defined in the context of our study (see our previous response).

C1-8 19 interannual evolution ‘of shoreline position’ – again need to be more precise

R1-8 We hope that it is now clearer after our rewording.

C1-9 25 “contrasted influence”?? what does this mean? – interacting influence? Improve.

R1-9 To make it clearer and more straightforward, the word “contrasted” was removed here.

C1-10 25 “3 main drivers” – change to ‘three main drivers’ – small numbers such as 3 should be stated as a word – three – throughout the manuscript.

R1-10 Corrected as suggested

C1-11 34-35 – what are the drivers of increased risk? Partly sea-level rise and climate change, but declining fluvial sediment input as in the Nile delta for example is also a huge effect and widespread?

R1-11 Indeed, on a long term (>decadal basis) scale, sea level rise and fluvial sedimentary influences become dominant, when compared with waves that are not projected to have a substantial increasing/decreasing trend worldwide (Melet et al., 2020).

New sentence was added Lines 38-39:

“On a decadal to centennial time scale, sea level rise and river influences will dominate, compared to waves that are projected have more contrasting, and small, increasing/decreasing trends globally. (Melet et al., 2020).”

C1-12 40 The appropriate SROCC and AR6 chapters would be good additional references here.

R1-12 We have added these chapter references now. Thank you.

C1-13 47-50 – The focus here is on physics-based models and only one driver?? Climate change – physics-based models should allow consideration of all relevant drivers, including climate change.

R1-13 To clarify this point on the physics-based versus simplified-physics models on a global basis, we have now added Lines 47-59:

“The advent of earth observation from space has greatly increased the availability of optical satellite data at global scale (Turner et al., 2021; Vitousek et al., 2022), which in combination with the computational power offered by cloud-based platforms (Traganos et al., 2018; Vos et al., 2019), have recently enabled global scale assessments of shoreline evolution (Luijendijk et al., 2018; Mentaschi et al., 2018) over the past three decades or so. Within this line of research, the linkages between observed shoreline changes and their potential dynamic drivers have yet to be analyzed in a comprehensive way to provide reliable projections of how the world’s shorelines may evolve in response to climate change (Ranasinghe 2016; Ranasinghe, 2020; Toimil et al., 2020). A comprehensive physics-based approach has yet to be developed globally. This is currently impossible due to the different scales at play at global and local scales (i.e., having an effect on the shoreline, such as wave breaking). A global application of existing ocean models coupled with waves-sea level-rivers-sediment (e.g., CROCO, Marchesiello et al., 2023, Delft3D, Roelvink et al., 1995) is out of reach. Thus, physically simplified shoreline models (such as CASCADE, Abessolo et al., 2021; Larson et al., 2003, COSMOS, Vitousek et al., 2017, ShorelineS, Roelvink et al., 2020, LX-Shore, Robinet et al., 2018; Tran et al., 2021) have a bright future as potential tools for investigating drivers of shoreline change at regional scales.”

Newly cited references:

Abessolo, G. O., Larson, M., & Almar, R. (2021). Modeling the Bight of Benin (Gulf of Guinea, West Africa) coastline response to natural and anthropogenic forcing. *Regional Studies in Marine Science*, 48, 101995.

Larson, M., Kraus, N. C., & Hanson, H. (2003). Simulation of regional longshore sediment transport and coastal evolution—the “Cascade” model. In *Coastal Engineering 2002: Solving Coastal Conundrums* (pp. 2612-2624).

Marchesiello, P., Chauchat, J., Shafiei, H., Almar, R., Benshila, R., Dumas, F., & Debreu, L. (2022). 3D wave-resolving simulation of sandbar migration. *Ocean Modelling*, 102127.

Robinet, A., Idier, D., Castelle, B., & Marieu, V. (2018). A reduced-complexity shoreline change model combining longshore and cross-shore processes: The LX-Shore model. *Environmental modelling & software*, 109, 1-16.

Roelvink, J. A., & Van Banning, G. K. F. M. (1995). Design and development of DELFT3D and application to coastal morphodynamics. *Oceanographic Literature Review*, 11(42), 925.

Roelvink, D., Huisman, B., Elghandour, A., Ghonim, M., & Reyns, J. (2020). Efficient modeling of complex sandy coastal evolution at monthly to century time scales. *Frontiers in Marine Science*, 7, 535.

Tran, Y. H., Marchesiello, P., Almar, R., Ho, D. T., Nguyen, T., Thuan, D. H., & Barthélemy, E. (2021). Combined longshore and cross-shore modeling for low-energy embayed sandy beaches. *Journal of Marine Science and Engineering*, 9(9), 979.

Vitousek, S., Barnard, P. L., Limber, P., Erikson, L., & Cole, B. (2017). A model integrating longshore and cross-shore processes for predicting long-term shoreline response to climate change. *Journal of Geophysical Research: Earth Surface*, 122(4), 782-806.

C1-14 303 – leading to how many points on the world’s coast? – 14,410 points (line 307)? Is this a good enough sample of the world’s coast as it is roughly a measurement every 100 km of shoreline length? Discussion of representativeness and scale is needed.

R1-14 Here we used a sampling resolution of 0.5° along the world's coastlines, which is about 50 km alongshore: so a total of 14,410 points spanning about 1.5 million kilometers (as in Almar et al., 2021, Nature Communications). This positions our study to capture the regional to global picture, but not the local scale (<50 km, ~ specific bay, beach, community seafront). Indeed, as mentioned above, the local scale comes with its own complexity (wave transformation on local bathymetry, impact of local infrastructure), which is beyond the reach and the scope of this study. Here we are more interested in finding similar regional behaviors than in distinguishing local diversity. Also, the study focuses on interannual evolution, which attenuates the complex linkages between drivers and shoreline evolution (with potential lags) found at, for example, monthly timescales (raw data here as our NDWI composite given by Google Earth Engine is monthly). To illustrate this

focus on both interannual and regional scales, Fig. S6 has been modified to include sites with longer term measurements, to show the differences between regional and local behavior, but also their similarity and the effects of aggregation of data on an annual basis.

Reference:

Almar, R., Ranasinghe, R., Bergsma, E. W., Diaz, H., Melet, A., Papa, F., ... & Kestenare, E. (2021). A global analysis of extreme coastal water levels with implications for potential coastal overtopping. *Nature communications*, 12(1), 3775.

C1-15 334 to 335 – is the same true for the Sand Engine in the Netherlands? What about the dynamic preservation of the whole coast of the Netherlands where the 1990 shoreline is maintained by law and nourishment is a process like the three factors considered in this paper? A comment is needed.

R1-15 We thank the Reviewer for highlighting this. Indeed, our approach considers natural stretches of coasts influenced by climate variability. This is now stressed in the limitations section. Influences other than climate variability on shoreline position has been discussed in a new paragraph in the limitations. Despite its relatively large influence, the Sand Engine spans about 3 km in the alongshore direction, and so it is out of scope and reach in our study with a 0.5° (~50 km) resolution regional to global study. Such an artificial outlier in our data would be smoothed out from the analysis by the regional aggregation of our data. Because all our data were detrended, the fact that a certain coastline is artificially maintained or not will not substantially affect our results which focuses on interannual evolution (or variability). Same for other kind of coastal infrastructures and management strategies (e.g., groins, nourishment, dikes, deep water harbor), which only have a local influence, not regional, and are smoothed out on a regional basis. Hopefully this question of scale and what is targeted and captured or not is now clearer in the revised manuscript.

C1-16 344 to 348 – sea level? So this is really climate-induced sea level and its variability – this needs to be stated as such. Many analyses include vertical land movement such as GIA which is excluded here. A clear statement of the implications and limitations is needed here.

R1-16 We thank the Reviewer for this important comment. Our study focuses on the interannual climate-induced variability. All signals are de-seasonalized and detrended (so no sea level rise here). Regional sea level used here is as observed by satellite altimetry, at about 25km from the coast. Its evolution is driven by ocean circulation and steric effect. While GIA and vertical land motion in general are important contributors to relative sea level rise, they are not driven by interannual climate variability. As such these effects were discarded from this study. Their effect would be seen in our data only in the shorelines trend, but because they are detrended, the influence of vertical land motion is neglected here. A discussion on the drivers of longer-term shoreline evolution is included in the Limitation section to further clarify the focus of this study.

C1-17 352 to 365 – run-off? Can you illustrate the method with an example of the Nile where the run-off to the Mediterranean is now essentially zero due to human use within the catchment. This is not uncommon in arid regions like the Indus or the Colorado in USA/Mexico. Human influence on run-off is profound in some large catchments – how is this represented in the model or is this not considered? This needs to be mentioned and may be a major uncertainty.

R1-17 In response to this comment, we have added additional details on the anthropogenic versus natural behavior of river discharge, runoff and sedimentation, as addressed in our study. The current version of global ISBA-CTRIP outputs used for river discharge does not take dams into account and therefore such perturbations are not included here. As a result, if a human influence were to result in regional (> 400 km) coastline changes, it would be captured in our coastline data. Such signals would be filtered out it appears as a trend, or retained if it contains w an interannual signature in the non-climate explained part of the shoreline variability in our analysis (climate modes and drivers do not explain 100% of the shoreline change signal in our data).

C1-18 641 to 642 – “large” to ‘Large’

R1-18 Corrected

C1-19 794 – why is the classic Duck station (not?) used? No worries on this point but it is an excellent dataset.

R1-19 We followed this advice and decided to use longer term surveyed dataset than what we had in the initial submission. As it was difficult to see the interannual evolution with such short (less than 3 years) data at Nha Trang (Vietnam) and Grand Popo (Benin), these sites were replaced by longer dataset of reference sites: Duck (US) and Narrabeen (Australia). To show the regional to local differences of shorelines but also their similarities, we show all our waterlines that were used to compute the regional average (used in our analysis) on monthly and yearly basis. Surveyed shoreline data are also aggregated to show their interannual behavior and verify if they match with our regional analysis. It is important to note that we compare here similar but still different features: waterlines (from satellite images) with shorelines (constant vertical reference), so we might expect more seasonal and interannual variability in the waterline due to sea level fluctuations. Surveyed data are local and reflect the local beach complexity (i.e., even more if bounded by cliffs) over a kilometer scale, and with the effects of natural local scale and human interventions such as nourishments. Our data are regional and smooth out all these local variations to enhance the regional evolution signal. However, it is striking in Fig. S6 that the satellite based and in-situ measured monthly evolutions are close and that the interannual evolutions are also similar. Again, it should be noted that our analysis does not focus on amplitude of change but focuses on spatio-temporal variability.

New Figure S6 looks:

[REDACTED]

Figure S6. Comparison of shorelines detected from satellite imagery using our auto-detection algorithm and in-situ measurements of shorelines at 4 locations around the world. a) Locations of the different sites from where in-situ measurements were used for the analysis; b) and c) Truc Vert, France; d) and e) Duck, USA; f) and g) Torrey Pines, USA and h) and i) Narrabeen, Australia. First column shows illustrations of Google earth images on which satellite derived shorelines at selected times are superimposed. The second column shows the in-situ data (colored lines) and the shorelines extracted from satellite images (black lines) at the closest computational coastal point. Thin lines show the monthly time series and thick lines the yearly time series. Grey shading indicates the geographical standard deviation within 400 km around the closest satellite-derived transect.

C1-20 844 – reference is incomplete

R1-20 We have double-checked and completed the references

Reviewer #2 :

C2-1 This is a study of how ENSO dominates year to year changes in coast lines over the period 2000-2017. The authors use a global dataset of satellite derived coast lines, plus measures of sea level variability, surface wave activity and river discharge, to show how the latter three are major controls on interannual coast line variability. They then look at measures of ENSO, SAM, and NAO variability to show that of these three, ENSO is the dominant global control as manifest through sea level, surface waves, and river discharge. This result makes sense, since ENSO is the only phenomenon used as a predictor of coast line variability that has a global fingerprint. The subject is interest, the paper is well written and organized, and the results are reasonably convincing. However, there are several questions that require clarification before publication.

R2-1 We thank the Reviewer for this positive assessment. We hope to have responded well to his/her constructive criticisms below.

C2-2 One is that the authors retain ENSO C-tones but ignore other regional modes of variability. In particular, based on only one reference, they ignore the IOD as largely ENSO driven. While IOD and ENSO are correlated, many other references could be marshaled to argue that dynamics internal to the Indian Ocean also play a role. That was obvious in the record breaking 2019 IOD that was associated with little if any ENSO forcing from the Pacific. Like the other regional modes that the authors do include, e.g., SAM and the NAO, the IOD will have a regional impact in the Indian Ocean. There are other regional modes with year-to-year variations as well that could be invoked, like the North Pacific Gyre Oscillation, the Atlantic interhemispheric gradient mode, etc. which regionally would account for as much if not more variability than the ENSO C-tones. So it strikes me as unbalanced if C-tones with their very small % variance explained are included and argued to be important, but not any of these other potentially influential climate modes.

R2-2 We thank the Reviewer for this interesting comment. First and foremost, in response to this comment, in the revised manuscript, we now focus on the two dominant modes of extratropical variability in each hemisphere, the Southern Annular Mode and the North Atlantic Oscillation (i.e, as the regional and oceanic expression of the Artic Annular Mode), as well as the dominant modes of tropical interannual climate variability in the Indian Ocean, the Indian Ocean Dipole, and in the Pacific, the El Niño Southern Oscillation, considered in all its spatial diversity and temporal complexity, in order to specifically account for its aforementioned seasonal effects on pantropical climate variability.

We would also likely note that the objective of this study is to evaluate the influences of different climate modes on shoreline interannual variability and its different hydrodynamic contributors. Such an approach has been adopted, for instance, by Roberts et al (2016) who used both observations and simulations of an eddy-permitting ocean model to evaluate the seasonal to decadal variability of dynamic sea level with respect to 15 modes of climate variability spanning the Atlantic, Pacific, Indian, and Southern Oceans. We deliberately restrain our focus here to only the most dominant basin-wide climate modes. Besides simplicity, this choice is also motivated by the fact that all the modes considered by Roberts et al. (2016) are not necessarily

independent from one another with some being potentially the regional expression of the dominant modes of climate variability at basin scale, which might hinder the ability to construct a simple climate-based statistical model for global interannual shoreline hindcasts, *i.e.*, the final objective of this study.

Indeed, a growing literature has recently begun to highlight that ENSO dominates both the Pacific-Indian Ocean (Jiang et al. 2021) and Pacific-Atlantic (Jiang et al. 2021, 2023) tropical interactions, shedding light on ENSO seasonal pacing (*i.e.*, combination mode) as a key predictor of interannual pantropical climate variability, and we now recognize this in our revised manuscript by also considering the IOD in our analysis, as suggested by the Reviewer.

We would also like to highlight here that, now in our revised manuscript, the ENSO combination tones account for a similar amount of explained variance as the other oceanic modes. This is mainly due to the fact that our statistical analysis is now carried out over a longer time period (1993-2019), which thus includes more ENSO events.

New text was added throughout our study, e.g. here Lines 83-86:

“Despite its strong dependence on ENSO (*Stuecker et al., 2017*), the Indian Ocean Dipole (IOD) can strongly influence climate variability in the Indian Ocean, especially in its western part, which is less influenced by the Pacific Ocean (*Marchant et al., 2007*), but also beyond the Indian basin (*Saji and Yamagata, 2003*).”

e.g. Lines 1993-1999:

“The distributions of correlation coefficients between observed and simulated (using Eq. 3) interannual anomalies of sea level, wave energy flux and river flows are shown in Figure 2a, b and c respectively. The model that accounts for the effect of the four climate modes (ENSO, IOD, NAO and SAM) produces an overall correlation of 0.72, 0.64, and 0.61 with the reanalysis products of SLA, wave energy, and river flows, respectively (*cf.* Table 1 of supplementary material), and exhibits statistically significant correlation at the 95% level along the world's shorelines. Figure 2d,e,f break down the respective contribution of NAO, IOD, SAM and ENSO to the model's globally averaged total variance.”

e.g. Lines 264-265:

“Similarly, the IOD variability unrelated to ENSO is mostly stochastic (*Stuecker et al., 2017*) and therefore of little value for improving seasonal climate predictions.”

And new Figure 2 looks:

Figure 2. Climate influence on drivers of shoreline change. Global distribution of correlation coefficients between observed and climate modes-based simulated (Equations 3) interannual anomalies of sea level (a.), wave energy (b.) and river discharges (c.). Respective percentage of global contributions of the different linear (E_{mode} and C_{mode}), non-linear (i.e. combination modes, E Comb-mode and C Comb-mode) ENSO terms, NAO, IOD and SAM to the total model solution for sea level (d.), wave energy (e.) and river discharges (f.). Gain in correlation between observed and simulated interannual anomalies of sea level (g.), wave energy flux (h.) and river flows (i.) respectively associated with the inclusion of NAO, IOD and SAM into the set of Equations (2). Whiskers in each inset delineate the range of one standard deviation among all randomized hindcasts of varying lengths from 10 to 27 years. In panels a), b) and c) only portions of shoreline where correlations are above the 95% confidence threshold are shown.

Newly cited Literature:

Roberts, C. D., Calvert, D., Dunstone, N., Hermanson, L., Palmer, M. D., & Smith, D., On the Drivers and Predictability of Seasonal-to-Interannual Variations in Regional Sea Level, *J. Climate*, 29(21), 7565-7585, (2016).

Jiang, F., Zhang, W., Jin, F. F., Stuecker, M. F., & Allan, R. (2021). El Niño Pacing Orchestrates Inter - Basin Pacific - Indian Ocean Interannual Connections. *Geophysical Research Letters*, 48(19), e2021GL095242.

Jiang, F., Zhang, W., Jin, F. F., & Stuecker, M. F. (2021). Meridional migration of ENSO impact on tropical Atlantic precipitation controlled by the seasonal cycle. *Geophysical Research Letters*, 48(24), e2021GL096365.

The following manuscript is currently in review and therefore not cited in the manuscript:

Jiang F., W, Zhang, F-F Jin, M.F. Stuecker, A. Timmermann, M.J. McPhaden, and J. Boucharel, Resolving the tropical Pacific/Atlantic interaction conundrum, submitted to *Proc. Natl. Acad. Sci.* (2023).

C2-3 The authors argue that including ENSO complexity is important. I agree, but comparisons of a “canonical” ENSO index (NINO3) and E- plus C- indices is cheating to a certain extent, since by now ENSO diversity/complexity is well known territory. A more interesting approach would be the following: NINO3 and the E-index are nearly identical; NINO4 and the C-index are also nearly identical (e.g., Cai et al, 2021, Nature Reviews, Earth and Environment). You can just as easily include ENSO complexity by using the NINO3 and NINO4 indices. In fact, there are advantages of using simple indices like NINO3 and NINO4 which do not require higher order mathematics. How would the results change if these indices were used instead of the E- and C- indices?

R2-3 We thank the reviewer for bringing up this interesting point. We agree that NINO3 and E-index and also NINO4 and C-index are respectively nearly identical and that they offer a simpler and more direct measure of ENSO spatial diversity, which indeed does not require an EOF analysis. As a matter of fact, the E and C indices developed by Ren and Jin (2011) are simply the rotated Niño3 and Niño4 indices, to make them independent of each other.

Reference:

Ren, H. L., & Jin, F. F. (2011). Niño indices for two types of ENSO. *Geophysical Research Letters*, 38(4).

As a consistency check, we re-made both Figure 2 and Figure 3 using the classical Niño indices, as suggested by the reviewer. The results are unchanged from our original versions, except that the variance explained by Niño3 is greater than that explained by the E index for all coastal drivers, which is likely due to the fact that the E index tends to capture more extreme events (such as the 1997/98 and 2015/16 ENSO events), whereas Niño3 captures more canonical ENSO variability.

Figure R1. Same as Figure 2 from the manuscript but using Niño3 and Niño4 indices instead of the E and C indices respectively.

Figure R2. Same as Figure 3 from the manuscript but using Niño3 and Niño4 indices instead of the E and C indices respectively.

Although the Niño3 and Niño4 indices are more readily available for seasonal predictions via climate centers for instance, we believe the use of the E and C indices is more mathematically sound and consistent with the recent literature on ENSO's seasonally modulated impacts on weather patterns and the climate system in general. However, we now include the results from the classical Niño indices in the supplementary material.

Newly cited Literature:

Ren, H.L, and F.F. Jin (2011), Niño indices for two types of ENSO, *Geophys. Res. Lett.*, 38.

C2-4 I do not profess to be a specialist in shoreline morphology and evolution, so I have what may appear to be a naive question. All the relationships between predictors and predictand (coast line) are presumed to be at zero lag on interannual time scales. However, intuitively, I would expect some of these predictors (e.g., surface waves) to have a cumulative or lagged effect. There should be more discussion of the assumption that everything varies in phase. There appears to be some of that in the section on degrees of freedom, but decorrelation time scale is not quite the same as lead-lag relationships.

R2-4 We thank the reviewer for this interesting remark. The cumulative effect of shoreline drivers on coastal morphology remains indeed an open question, and has not been well addressed in the literature because of the need of long time series. However, because our estimation of the shoreline from space is based on the water line position, we expect an instantaneous response of shoreline to SLA variation. The study by Angnuureng et al. (2017) shows for instance that the memory of shoreline with respect to the wave forcing does not exceed two weeks. Similar findings have been reported for Australian beaches by Ranasinghe et al. (2012). Although the time scales of shoreline response to the input of sediments from river discharge remains somewhat uncertain, a local lead-lag correlation analysis between the shoreline variability and its dominant drivers shows in the new Figure S8 that the maximum coherency always occurs at a lag below 12 months. Since we explore the interannual variability of the shoreline at regional scale, this implies an in-phase relationship between its position and all hydrodynamic forcing. We have now stated this in the Limitations sections of the revised manuscript as follows (Lines 504-516):

“Statistical significance of correlations and intervals of confidence

Here, we consider a total period analysis spanning 324 months but with a temporal smoothing using an 8-months window running mean, which leads to $324/8 = 40$ independent time steps. In Figure S5, we estimated the slope of the best linear fit to the autocorrelation logarithm for each driver and the shoreline, or in other terms, an e-folding time equivalent to their interannual memory. While such memory times vary differently for the drivers and the shoreline position along the different global shorelines, they remain below a maximum of ~24 months. In addition, because our estimation of the shoreline from space is based on the waterline position, we expect an instantaneous response of shoreline to SLA variation. The studies by Ranasinghe et al. 2012 in Australia and Angnuureng et al. 2017 in France show with high frequency observations that the memory of

shoreline with respect to the wave forcing does not exceed two weeks. Although the time scales of the shoreline's response to the input of sediments from river discharge remains somewhat uncertain, a local lead-lag correlation analysis between the shoreline variability and its dominant driver's shows that the maximum coherency always occurs at a lag below 12 months (Figure S8). Since we explore the interannual variability of the shoreline, this implies an in-phase relationship between its position and all hydrodynamic forcing."

New Figure S8 looks:

Figure S8. Local lead-lag analysis (months of the maximum correlation between shoreline and its dominant drivers interannual variability).

Newly cited Literature:

Angnuureng, D. B., Almar, R., Senechal, N., Castelle, B., Addo, K. A., Marieu, V., & Ranasinghe, R. (2017). Shoreline resilience to individual storms and storm clusters on a meso-macrotidal barred beach. *Geomorphology*, 290, 265-276.

Karunaratna, H., Pender, D., Ranasinghe, R., Short, A. D., & Reeve, D. E. (2014). The effects of storm clustering on beach profile variability. *Marine geology*, 348, 103-112.

Ranasinghe, R., Holman, R., de Schipper, M. A., Lippmann, T., Wehof, J., Duong, T., Roelvink, D. and Stive, M.J.F., 2012. Quantifying morphological recovery time scales using Argus video imaging: Palm Beach, Sydney and Duck, NC. *Proceedings of the 33rd International Conference on Coastal Engineering (ICCE) 2012, Santander, Spain*.

Other:

C2-5 *The figures have terrible resolution and are in some cases impossible to read.

R2-5 We apologize for the poor resolution of the figures in the original manuscript. We have updated the resolution of all Figures in the revised version.

C2-6 *The authors incorrectly state that the E-index and C-index are uncorrelated. The first and second EOFs of SST are uncorrelated. Their linear combinations in the form of the E- and C-indices are not.

R2-6 Indeed, as the first and second Principal components of the SST decomposition into EOFs, the E and C indices are indeed uncorrelated and therefore offer independent predictors in our multi-linear regression analysis. However, we agree that the other climate modes and the ENSO combination modes are not uncorrelated. We have changed this incorrect statement and added explanatory text in the revised manuscript (Lines 455-463) that now reads:

“ E_{mode} and C_{mode} are two uncorrelated and independent ENSO indices, calculated as the first two rotated Principal Components of the EOF decomposition of SST interannual anomalies (Takahashi et al., 2011) and accounting for the variability of the two different types of ENSO, respectively the extreme warm events in the Eastern (i.e. EP El Niño) and moderate warm events in the Central Pacific (i.e. CP El Niño). Note that the two classical ENSO indices Niño3 (monthly sea surface temperature anomalies averaged in the region bounded by 5°N to 5°S, from 170°W to 120°W) and Niño4 (monthly sea surface temperature anomalies averaged in the region bounded by 5°N to 5°S, from 150°W to 90°W) are almost identical to the E_{mode} and C_{mode} indices, respectively, and provide a similar but simpler and more direct measure of ENSO spatial diversity, although not quite orthogonal. Nevertheless, we re-ran all the main calculations and figures using these simple indices, which gave similar results (see Figures S5 and S6).”

C2-7 *In regression analysis, degrees of freedom should be adjusted for the number of predictors. I did not see any discussion of this topic and how it affect the uncertainty of the estimated parameters.

R2-7 We agree with the reviewer and apologize for our crude approximation. We have now updated the correlation thresholds for statistical significance accounting for the numbers of predictors in the multi-linear regression analysis (Lines 504-521):

“Statistical significance of correlations and intervals of confidence

Here, we consider a total period analysis spanning 324 months but with a temporal smoothing using an 8-months window running mean, which leads to $324/8 = 40$ independent time steps. In Figure S5, we estimated the slope of the best linear fit to the autocorrelation logarithm for each driver and the shoreline, or in other terms, an e-folding time equivalent to their interannual memory. While such memory times vary differently for

the drivers and the shoreline position along the different global shorelines, they remain below a maximum of ~24 months. In addition, because our estimation of the shoreline from space is based on the waterline position, we expect an instantaneous response of shoreline to SLA variation. The studies by Ranasinghe et al. 2012 in Australia and Angnuureng et al. 2017 in France show with high frequency observations that the memory of shoreline with respect to the wave forcing does not exceed two weeks. Although the time scales of the shoreline's response to the input of sediments from river discharge remains somewhat uncertain, a local lead-lag correlation analysis between the shoreline variability and its dominant driver's shows that the maximum coherency always occurs at a lag below 12 months (Figure S8). Since we explore the interannual variability of the shoreline, this implies an in-phase relationship between its position and all hydrodynamic forcing.

This leads to a total number of independent observations for shoreline and drivers of $N = 40 - (24/8) = 37$. Thus, the total number of degrees of freedom d.o.f for a multiple linear regression analysis with k predictors is $d.o.f = N - k - 1$. For instance, this gives a $d.o.f = 37 - 3 - 1 = 33$ and correlation coefficient thresholds of 0.32 and 0.44 at the 95% and 99% significance level respectively according to a Student t-test for Figure 1. For Figure 2, $d.o.f = 37 - 7 - 1 = 29$ and correlation coefficient thresholds are 0.36 and 0.46 respectively. "

C2-8 *How accurate are individual estimates of shoreline position from space? The year to year changes as shown in Figure S6 are ± 5 -10 m, which is not large, so the noise to signal must be significant.

R2-8 We thank the Reviewer for raising this important point. The spatial resolution of raw NDWI monthly composite maps for Landsat is 30 m. Data over a buffer zone around each transect are projected on the shore normal transect, thus resampling the data to a much finer resolution (submetric). However, several processes affect the shoreline accuracy, such as tides (not resolved in this monthly dataset of NDWI) and wave-induced swash (whose phase is impossible to determine from satellite imagery) that are inherent issues in satellite derived shorelines (see Vos et al., 2019) and most of these issues are addressed in short term and local studies using individual shoreline images rather than monthly composites. However, using such individual images for each site is impossible in our global study. Instead, we maintain larger scale accuracy by aggregating the data spatially at a regional scale (~400 km) and temporally (annually). To address our ability to capture interannual variability, the new Fig. S6 shows the regional dispersion from coast to coast and the general behavior, both at monthly and interannual scales, together with in-situ data. Please note that, as suggested by Reviewer 1, we have changed the comparison locations (from those used in the previous manuscript) to sites with longer in-situ measurement datasets. Overall, Figure S6 shows that the regional shorelines evolve similarly and have a similar interannual imprint.

This is discussed throughout the manuscript and particularly in Data and methods (Shorelines from satellite images sub section) (Lines 365-386):

"The global dataset used in this study is re-sampled with transects spaced at 0.5° intervals (~50 km), along the same 14,410 points vector as in *Almar et al. (2021)* spanning approximately 1.5 million kilometers. The initial

shoreline dataset used is Global Self-consistent Hierarchical High-resolution Geography (GSHHG version 2.3.6, Wessel and Smith, 1996) to define locations along the world shorelines. The world was divided into computational regions using a series of GSHHG shoreline polygons. This positions our study not at the local scale (<50 km, ~ specific bay, beach, community seafront), but to capture the regional to global picture. The local coastline has its own complexities (e.g., wave transformation on unknown changing bathymetry, impact of infrastructure and human intervention), which are beyond the scope of this study. Instead, the individual data are more regionally aggregated (along 8 consecutive data points, ~400 km of coastline), showing similar regional behavior rather than distinguishing local diversity (see Fig. S4). The monthly composites of shoreline positions were derived from 1993 to 2019 using multiple satellite acquisitions provided by the Landsat missions 5, 7 and 8. The extraction of these data was performed on the Google Earth Engine (GEE) platform (*Gorelick et al., 2017*). The GEE was calculated over 30 regions of interest of varying size, covering coastal areas worldwide (and 60% of the globe). Since we used T1_8DAY_NDWI 30 m collections from Landsat 5, 7 and 8 satellites, which represent monthly median composites of 10.5 images (i.e., 3 or 4 images per month depending on the month x 3 satellites) of size 0.70 x global area at a 30 m resolution, about 400 Megapixels were processed, which amounts to approximately three petaoctets of satellite data and required 7200 hours of computation. Normal Difference Water Index (NDWI) maps were derived from satellite images and the NDWI threshold used was 0 (*McFeeters, 1996*). The identified pixels correspond to ocean for $NDWI > 0$, and to land surfaces for $NDWI < 0$. The shoreline is then identified as the interface between the land and sea surfaces (*Kelly and Gontz, 2018*). We acknowledge that the selection of water index thresholds can have a significant impact on the quality and distribution of satellite-derived shorelines. The constant NDWI threshold used here contrasts with the use of more complex dynamic methods to optimize thresholds to local conditions (e.g., the commonly used Otsu method; *Otsu, 1979*), but remains the most commonly used approach to obtain a primary estimate and gives reasonable results at the validation sites.”

New Figure S6 looks:

[REDACTED]

Figure S6. Comparison of shorelines detected from satellite imagery using our auto-detection algorithm and in-situ measurements of shorelines at 4 locations around the world. a): Locations of the different sites from where in-situ measurements were used for the analysis; b) and c) Truc Vert, France; d) and e) Duck, USA; f) and g) Torrey Pines, USA and h) and i) Narrabeen, Australia. First column shows illustrations of Google earth images on which satellite derived shorelines at selected times are superimposed. The second column shows the in-situ data (colored lines) and the shorelines extracted from satellite images (black lines) at the closest computational coastal point. Thin lines show the monthly time series and thick lines the yearly time series. Grey shading indicates the geographical standard deviation within 400 km around the closest satellite-derived transect.

Reviewer #3 :

C3-1 I have reviewed the above manuscript and find that it will be suitable for publication in Nature Communications following moderate revisions. The manuscript is very well written and therefore I am not submitting an annotated version of the manuscript. Below please find general and specific comments concerning possible suggestions for the authors to address.

R3-1 We thank the Reviewer for this positive assessment of our manuscript. We greatly appreciate his/her constructive comments, which have helped us to improve the manuscript.

General Comments:

The paper “The influence of El Nino on global shorelines”, presents a global data set of satellite derived shorelines, a conceptual model of the primary drivers of the shoreline evolution captured by the data, and a quantitative framework exploring the role of ENSO variability in forcing their interannual variability in both the drivers and the morphological response.

C3-2 My first general comment is regarding the uncertainty of the satellite derived shorelines (SDS). Recent studies have shown that the uncertainty of SDS varies with shoreline type and metocean forcing conditions and therefore must vary widely globally and with time (satellite mission). A more quantitative handling of SDS uncertainty is warranted. For example, the four sites chosen to assess the quality of the SDS data all are areas that are not subjected to persistent cloud cover. Most likely SDS uncertainty in regions that do have significant cloud cover would be much worse. Further, the model developed in this paper attempts to explain the detrended monthly shoreline anomaly – with sub annual dynamics explicitly removed. The ‘good agreement’ on the SDS quality plots shown in Figure S6 have reasonable correlation precisely because the seasonal signal is left in. It is not clear, at least to me, how well the filtered SDS data (8-months window running mean) would recover the similarly filtered shoreline signal from in situ data.

R3-2 We thank the reviewer for this suggestion to be more precise regarding the validation of our SDS. Indeed, it is important to clearly assess the potential and limitations of our global SDS dataset, to position our work in the broad context of other similar studies. There are numerous local to regional studies that target an accurate SDS, for instance on individual images (such as Vos et al., 2023). Here we used a constant NDWI threshold value on monthly composites, worldwide, ending up with a 0.5° (~50 km) resolution with 8-months smoothed (or yearly-aggregated monthly anomalies for the time series presented in Figure 3) SDS and shoreline drivers dataset: in line with our target to position our study as a regional to global first pass study on the influence of

main shoreline drivers and climate modes on shoreline evolution, at interannual scales. Also, tide, wave-induced swash, human impacts (e.g., nourishment, infrastructures) have a large influence on local and short timescales which are dampened by our spatial and temporal aggregation/smoothing. Also, here we used offshore waves and regional sea levels, the characteristics of which can undergo complex transformations before reaching the coast, including phase and amplitude. Despite the computational efficiency of our approach, these omissions necessarily limit our study to interannual time scales and regional to global spatial patterns, the local scale being out of scope and also clearly out of reach.

To better illustrate how we position our study, and following the reviewer's recommendations, a new Figure S6 has been produced, replacing the reference sites with those that have longer in-situ shoreline datasets: Truc Vert (Atlantic SW coast of France), Narrabeen (Australia East Coast), Duck (USA East coast) and Torrey Pines (USA West coast). All the SDS within our regional aggregation are displayed to show the differences and also their similarities. As suggested by the Reviewer, the data are also processed with the 8-month window running mean to better capture the interannual behavior, which is in fact our target time scale. This detailed in the revised text (Lines 390-413):

“To illustrate the ability but also the limitations of our method to observe shoreline variability from satellite, Figure S6 shows a comparison between the closest satellite transects and various ground measurements of some of the longest shoreline datasets around the world: Truc Vert (South West France, Figure S6a, from *Castelle et al., 2020*), Torrey Pines (West Coast USA, Figure S6b, from *Ludka et al., 2019*), Duck (East coast USA, Figure S6c, provided by the U.S. Army Engineer Research & Development Centre, Coast & Hydraulics Laboratory, Field Research Facility) and Narrabeen (East Coast Australia, Figure S6d, *Turner et al., 2016*). The in-situ data are based on regular monthly topo-bathymetry measurements averaged along the coast (typically one kilometre), and the comparative shoreline proxy used here is the high tide upper beach contour above mean sea level. For all sites, the ground truth data are interpolated to a regular monthly resolution, and comparisons are made for periods where no significant gaps were present in the in-situ data. Despite the coarse resolution of our dataset (transects every 0.5°), our regional comparison with local measurements shows good overall agreement, increasing from short, seasonal, to longer interannual time scales. The local behavior of the beaches such as a nourishment at Torrey Pines cannot be captured and is beyond the scope of our regional to global analysis. Here, we are after the variability of the shoreline for which the correlation is the most appropriate quality proxy. The correlation is used to assess the quality of our dataset compared to in-situ data. It should be emphasized that we aim here to resolve only the regional to global scales of interannual variability of the coastal shoreline, not the amplitude of the subsequent numerous and diverse processes that may include non-linearities and interactions within the coastal system (*Anthony and Aagaard, 2020; Bergsma et al., 2022*). These correlation coefficients between our satellite-derived shorelines representative of the regional scale, and in-situ local shorelines range from 0.38 to 0.61 at these sites, despite distances of up to tens of kilometers between our closest transects and the sites. The differences may come from the difference in the shoreline approximation used; thus, all sea level variations, such as regional sea level, wave contribution to sea level at the coast (i.e., setup and run-up) but also river discharge have a more

direct effect on the position of the waterline than the surveyed shoreline proxy using mean sea level as a reference. Nevertheless, this demonstrates the regional common behavior of shorelines at interannual scales, already identified (e.g., *Barnard et al. 2016; Masselink et al., 2014; Vos et al., 2023*).“

And the new Figure S6 looks :

[REDACTED]

Figure S6. Comparison of shorelines detected from satellite imagery using our auto-detection algorithm and in-situ measurements of shorelines at 4 locations around the world. a): Locations of the different sites from where in-situ

measurements were used for the analysis; b) and c) Truc Vert, France; d) and e) Duck, USA; f) and g) Torrey Pines, USA and h) and i) Narrabeen, Australia. First column shows illustrations of Google earth images on which satellite derived shorelines at selected times are superimposed. The second column shows the in-situ data (colored lines) and the shorelines extracted from satellite images (black lines) at the closest computational coastal point. Thin lines show the monthly time series and thick lines the yearly time series. Grey shading indicates the geographical standard deviation within 400 km around the closest satellite-derived transect.

C3-3 I am also concerned that perhaps the abstract oversells the paper's contribution. The statement: 'Our results represent a new framework for understanding and predicting climate driven coastal hazards' is odd considering how much work has made to identify a direct connection between ENSO and coastal change over the last several decades. In fact, a general comment is that the paper lacks a thorough set of citations regarding the influence of ENSO on shorelines. At present there are more citations focused on the role of the NAO on shoreline variability than citations listed focused on the already demonstrated role of ENSO.

R3-3 In response to this comment, more references on the influence of ENSO on shorelines have now been added to the revised manuscript. We hope we now better acknowledge these key previous works. In particular text was added Lines 80-86:

"While ENSO is known to be unambiguously dominant in the Pacific with strong impacts on the shoreline and coastal ecosystems (Storlazzi and Griggs, 1998; Ranasinghe et al., 2004; Biribo and Woodroffe, 2013; Barnard et al., 2011, 2015, 2017; Harley et al., 2017; Young et al., 2018; Cuttler et al., 2020; Duke et al., 2022; Vos et al., 2023), other climate modes can also significantly modulate coastal drivers in other ocean basins. Despite its strong dependence on ENSO (Stuecker et al., 2017), the Indian Ocean Dipole (IOD) can strongly influence climate variability in the Indian Ocean, especially in its western part, which is less influenced by the Pacific Ocean (Marchant et al., 2007), but also beyond the Indian basin (Saji and Yamagata, 2003)."

Newly cited Literature:

Duke NC, Mackenzie JR, Canning AD, Hutley LB, Bourke AJ, Kovacs JM, et al. (2022) ENSO-driven extreme oscillations in mean sea level destabilise critical shoreline mangroves—An emerging threat. *PLOS Clim* 1(8): e0000037. <https://doi.org/10.1371/journal.pclm.0000037>

Storlazzi, Curt D., and Gary B. Griggs. "Influence of El Niño-Southern Oscillation (ENSO) Events on the Coastline of Central California." *Journal of Coastal Research*, 1998, 146–53. <http://www.jstor.org/stable/25736131>.

Vos, K., Harley, M. D., Turner, I. L., & Splinter, K. D. (2023). Pacific shoreline erosion and accretion patterns controlled by El Niño/Southern Oscillation. *Nature Geoscience*, 16(2), 140-146.

Biribo, N., & Woodroffe, C. D. (2013). Historical area and shoreline change of reef islands around Tarawa Atoll, Kiribati. *Sustainability Science*, 8, 345-362.

Mortlock, Thomas R., and Ian D. Goodwin. "Impacts of enhanced central Pacific ENSO on wave climate and headland-bay beach morphology." *Continental Shelf Research* 120 (2016): 14-25.

Barnard, P. L. et al. The impact of the 2009–10 El Niño Modoki on US West Coast beaches. *Geophys. Res. Lett.* <https://doi.org/10.1029/2011GL047707> (2011).

Barnard, P. L. et al. Coastal vulnerability across the Pacific dominated by El Niño/Southern Oscillation. *Nat. Geosci.* **8**, 801–807 (2015).

Barnard, P. L. et al. Extreme oceanographic forcing and coastal response due to the 2015–2016 El Niño. *Nat. Commun.* **8**, 14365 (2017).

Ranasinghe, R., McLoughlin, R., Short, A. & Symonds, G. The Southern Oscillation Index, wave climate, and beach rotation. *Mar. Geol.* **204**, 273–287 (2004).

Young, A. P. et al. Southern California coastal response to the 2015–2016 El Niño. *J. Geophys. Res. Earth Surf.* **123**, 3069–3083 (2018).

Cuttler, M. V., Vos, K., Branson, P., Hansen, J. E., O’leary, M., Browne, N. K., & Lowe, R. J. (2020). Interannual response of reef islands to climate-driven variations in water level and wave climate. *Remote Sensing*, *12*(24), 4089.

C3-4 The remainder of the paper is quite strong, and the approach presented for disentangling the relative influence of the various climate driver flavors is a significant contribution.

R3-4 We thank the reviewer for this very positive assessment.

Specific Comments on Text:

C3-5 Line 47-50: This is a personal preference of mine, but why present these statements in a negative light (using words like ‘not’, and ‘prevents’...)? The historical global assessments that you are referring to are a huge, very recent advance and provide the opportunity for the analyses you are presenting with this paper. It seems easy enough to frame your contribution as an evolution in knowledge rather than simply filling in what others did not think to do.

R3-5 Following the Reviewer advice, we have now rephrased this sentence in a more positive way (Lines 47-53):

“The advent of earth observation from space has greatly increased the availability of optical satellite data at global scale (Turner et al., 2021; Vitousek et al., 2022), which in combination with the computational power offered by cloud-based platforms (Traganos et al., 2018; Vos et al., 2019), have recently enabled global scale assessments of shoreline evolution (Luijendijk et al., 2018; Mentaschi et al., 2018) over the past three decades or so. Within this line of research, the linkages between observed shoreline changes and their potential dynamic drivers have yet to be analyzed in a comprehensive way to provide reliable projections of how the world’s shorelines may evolve in response to climate change (Ranasinghe 2016; Ranasinghe, 2020; Toimil et al, 2020). “

C3-6 Line 54: Oddly worded. Certainly hundreds (perhaps thousands) of studies have ‘highlighted the contributions of wind-generated ocean surface waves...’ to ‘erosion’.

R3-6 Indeed the Reviewer is correct, we have now changed the text accordingly (Lines 65-68):

“However, while wind-generated ocean surface waves are known to dominate the impact on beaches at short event to sub-annual scales (e.g. *Bergsma et al., 2022; Montañó et al., 2020*), recent studies have highlighted the contribution of waves to longer-term interannual coastal water levels (*Melet et al., 2018; Kirezci et al., 2020; Almar et al., 2021*) and erosion (e.g. *Barnard et al., 2015; 2017; Mentaschi et al., 2018; Vos et al., 2023*).”

C3-7 Line 131: Change evolution to variability??

R3-7 Thanks to this comment, we decided to not use the term evolution but rather change or variability throughout the manuscript, as here we rather focus on the temporal variability of the (normalized) shoreline position rather than an actual position evolution with an amplitude.

Title was also changed to a more precise formulation:

“Influence of El Niño on the variability of global shoreline position”

C3-8 Line 136: Change ‘governs’ to ‘govern’.

R3-8 This word does not appear any more in the revised manuscript.

C3-9 Eq 3: better to not reuse gamma here?

R3-9 We thank the Reviewer for this good suggestion. We have now changed this term in the new Equation 3 (Line 192):

$$\begin{cases} \text{Sea level}(x, t) = f_1(ENSO) + \varphi_1(x)NAO + \delta_1(x)SAM + \rho_1(x)IOD \\ \text{Wave energy}(x, t) = f_2(ENSO) + \varphi_2(x)NAO + \delta_2(x)SAM + \rho_2(x)IOD \\ \text{River flow}(x, t) = f_3(ENSO) + \varphi_3(x)NAO + \delta_3(x)SAM + \rho_3(x)IOD \end{cases} \quad (3)$$

C3-10 Lines 264-265: What about the possibility that the uncertainties in satellite derived shorelines are actually responsible for the significant relationships?

R3-10 We thank the Reviewer for this constructive comment, which was also stressed in his/her general comments. We hope the limitations and uncertainty linked to the SDS are now better quantified and detailed in the revised manuscript. In particular, here the NDWI waterline is used as a proxy of the shoreline, which is now more clearly stated from one of the first section of the manuscript Drivers of shoreline change (Lines 112-115):

“We use the waterline as a proxy for shoreline. Since our focus here is on the interannual variability of climate-driven variability at the global scale, we consider the variability of the monthly drivers smoothed with an 8-month window running mean (to remove sub-annual dynamics from our analysis) and through simplified expressions of their dominant contributors.”

As such, any fluctuations of the sea level, such as the altimetry regional sea level, wave contribution to sea level at the coast (i.e., setup and runup) but also the river run off have a direct influence on the waterline position, more so than a shoreline proxy that takes the mean sea level as a reference would. In any case, because of our annual and regional aggregation, we do not expect an uncertainty being responsible for significant relationships we have observed. This also now deeply discussed in Data and Methods Section (Lines 357-413):

“Shorelines from satellite images

Here we use the water line as shoreline definition (*Boak and Turner, 2005*), i.e., the water line at the time of data collection. Due to the continuous influence of tides, storm surges and waves on the shoreline, the water line is subject to a combination of sediment and hydrodynamic variabilities that do not directly represent the evolution of the “geological” shoreline, such as the retreat of mean high-water line, the vegetation line, the erosion of a cliff, or the erosion of a coastal settlement. Different portions of the shoreface profile are likely to have contrasting responses to drivers of change, even potentially exhibiting contrasting trends through time and space (e.g., *Castelle et al., 2014; Cowley et al. 2022*). Nevertheless, the water line adequately reflects the shoreline position that is relevant for vulnerability and risk associated with erosion and flooding (*Bishop-Taylor et al. 2021*) and is thus used a shoreline proxy in this study.

The global dataset used in this study is re-sampled with transects spaced at 0.5° intervals (~ 50 km), along the same 14,410 points vector as in *Almar et al. (2021)* spanning approximately 1.5 million kilometers. The initial shoreline dataset used is Global Self-consistent Hierarchical High-resolution Geography (GSHHG version 2.3.6, Wessel and Smith, 1996) to define locations along the world shorelines. The world was divided into computational regions using a series of GSHHG shoreline polygons. This positions our study not at the local scale (<50 km, \sim specific bay, beach, community seafront), but to capture the regional to global picture. The local coastline has its own complexities (e.g., wave transformation on unknown changing bathymetry, impact of infrastructure and human intervention), which are beyond the scope of this study. Instead, the individual data are more regionally aggregated (along 8 consecutive data points, ~ 400 km of coastline), showing similar regional behavior rather than distinguishing local diversity (see Fig. S4). The monthly composites of shoreline positions were derived from 1993 to 2019 using multiple satellite acquisitions provided by the Landsat missions 5, 7 and 8. The extraction of these data was performed on the Google Earth Engine (GEE) platform (*Gorelick et al., 2017*). The GEE was calculated over 30 regions of interest of varying size, covering coastal areas worldwide (and 60% of the globe). Since we used T1_8DAY_NDWI 30 m collections from Landsat 5, 7 and 8 satellites, which represent monthly median composites of 10.5 images (i.e., 3 or 4 images per month depending on the month x 3 satellites) of size 0.70 x global area at a 30 m resolution, about 400 Megapixels were processed, which amounts to approximately three petaoctets of satellite data and required 7200 hours of computation. Normal Difference Water Index (NDWI) maps were derived from satellite images and the NDWI threshold used was 0 (*McFeeters, 1996*). The identified pixels correspond to ocean for $NDWI > 0$, and to land surfaces for $NDWI < 0$. The shoreline is then identified as the interface between the land and sea surfaces (*Kelly and Gontz, 2018*). We acknowledge that the selection of water index thresholds can have a significant impact on the quality and distribution of satellite-derived shorelines. The constant NDWI threshold used here contrasts with

the use of more complex dynamic methods to optimize thresholds to local conditions (e.g., the commonly used Otsu method; *Otsu, 1979*), but remains the most commonly used approach to obtain a primary estimate and gives reasonable results at the validation sites.

Issues due to wave breaking or water turbidity during extremes are smoothed out using monthly median composites in addition to the post 8-month smoothing to remove event-related and sub-annual dynamic. Also, our study focuses on interannual evolution, which dampens the complexity of this short-term link between drivers and shoreline evolution (with potential lags, e.g., *Angnuureng et al., 2017; Karunarathna et al., 2014*). To illustrate the ability but also the limitations of our method to observe shoreline variability from satellite, Figure S6 shows a comparison between the closest satellite transects and various ground measurement of some of the longest shoreline datasets around the world: Truc Vert (South West France, Figure S6a, from *Castelle et al., 2020*), Torrey Pines (West Coast USA, Figure S6b, from *Ludka et al., 2019*), Duck (East coast USA, Figure S6c, provided by the U.S. Army Engineer Research & Development Centre, Coast & Hydraulics Laboratory, Field Research Facility) and Narrabeen (East Coast Australia, Figure S6d, *Turner et al., 2016*). The in-situ data are based on regular monthly topo-bathymetry measurements averaged along the coast (typically one kilometre), and the comparative shoreline proxy used here is the high tide upper beach contour above mean sea level. For all sites, the ground truth data are interpolated to a regular monthly resolution, and comparisons are made for periods where no significant gaps were present in the in-situ data. Despite the coarse resolution of our dataset (transects every 0.5°), our regional comparison with local measurements shows good overall agreement, increasing from short, seasonal, to longer interannual time scales. The local behavior of the beaches such as a nourishment at Torrey Pines cannot be captured and is beyond the scope of our regional to global analysis. Here, we are after the variability of the shoreline for which the correlation is the most appropriate quality proxy. The correlation is used to assess the quality of our dataset compared to in-situ data. It should be emphasized that we aim here to resolve only the regional to global scales of interannual variability of the coastal shoreline, not the amplitude of the subsequent numerous and diverse processes that may include non-linearities and interactions within the coastal system (*Anthony and Aagaard, 2020; Bergsma et al., 2022*). These correlation coefficients between our satellite-derived shorelines representative of the regional scale, and in-situ local shorelines range from 0.38 to 0.61 at these sites, despite distances of up to tens of kilometers between our closest transects and the sites. The differences may come from the difference in the shoreline approximation used; thus, all sea level variations, such as regional sea level, wave contribution to sea level at the coast (i.e., setup and run-up) but also river discharge have a more direct effect on the position of the waterline than the surveyed shoreline proxy using mean sea level as a reference. Nevertheless, this demonstrates the regional common behavior of shorelines at interannual scales, already identified (e.g., *Barnard et al. 2016; Masselink et al., 2014; Vos et al., 2023*).“

But also the uncertainties linked to the satellite-derived shorelines used here are discussed in the Limitations section (Lines 336-350):

“Satellite-derived shorelines can be prone to many sources of uncertainties or systematic biases that can confound analyses such as those presented here. Therefore, it remains challenging to assess whether the absence of a relationship with potential drivers (e.g., ENSO) is due to a true lack of relationship, or simply due to poor quality shoreline data. The monthly median NDWI shoreline mapping approach used here is likely to be more susceptible to potential data issues than other approaches that use longer annual composites (e.g. Luijendijk et al. 2018, Bishop-Taylor et al. 2021), particularly in regions of the world with either high persistent cloud cover, or relatively low satellite observation densities (Wulder et al. 2016; Bersgma et al., 2020). This can make it challenging to obtain even clean annual median shorelines in many of these low data environments, let alone high-quality monthly shorelines. Similarly to a former global study by Luijendijk et al., (2018) and unlike Vos et al., (2019) who used advanced trained convolutional neural network coefficients to distinguish between land and marine pixels, here we used the more basic NDWI waterline proxy for this global application. Our shorelines are smoothed in the same way as the drivers over an 8-month period to eliminate sub-annual shoreline dynamics, which also smooths out some of the problems associated with getting good monthly shorelines. The correlations obtained between our independent drivers/climate modes and the observed/modelled shoreline changes provide a reasonable level of confidence in our satellite-based global shoreline dataset that admittedly can be further improved, but paves the way for more future detailed studies and technological developments.”

C3-11 Line 274: change to ‘smooths’

R3-11 Done

C3-12 Line 278-285: Not mentioning beach nourishment in this discussion seems like a significant oversight.

R3-12 We thank the Reviewer but the case of engineered beach is mentioned in the Limitations section (Lines 324-334):

“Furthermore, our methodology considers natural stretches of coasts influenced by natural climate variability. However, shorelines have been actually modified in various ways by human activities, particularly in urbanized areas where, for example harbors have been constructed, land reclaimed from the ocean (Luijendijk, et al., 2018), seawalls built to combat shoreline recession, cliffs stabilized, beaches nourished, and groins placed in an attempt to retain a beach fringe and maintain dunes. For example, in the US alone, 14% of national shoreline is estimated to be hardened with engineering structures (e.g. seawalls, dikes - Gittman et al., 2015), and this percentage is expected to intensify globally over the 21s century (Cao et al., 2021; Floerl et al., 2021). Human intervention is particularly high in tropical developing countries, where dramatic changes in land use are occurring, due to, for instance, deforestation and urbanization, at a higher rate than anywhere else in the world (Alves et al., 2020; Dada et al., 2021). In particular, unplanned or poorly designed coastal structures are a major issue transforming the coastal landscape in these countries. The regional aggregation of our data at

synoptical scale (8 transects, ~400 km) aims at damping these local and human-induced influences to enhance and capture larger scales climate-driven patterns. “

C3-13 Line 314: What do you mean by OTSU? Please elaborate or provide a citation.

R3-13 Indeed this was not clear enough. A reference is now (Otsu, 1979) provided for the commonly used Otsu method of automatic thresholding based on the aspect of the histogram. New text reads (Lines 384-386):

“The constant NDWI threshold used here contrasts with the use of more complex dynamic methods to optimize thresholds to local conditions (e.g., the commonly used Otsu method; Otsu, 1979), but remains the most commonly used approach to obtain a primary estimate and gives reasonable results at the validation sites.”

A referenced was added:

Otsu, Nobuyuki. « A threshold selection method from gray-level histograms », IEEE Trans. Sys., Man., Cyber., vol. 9, 1979, p. 62–66 (DOI 10.1109/TSMC.1979.4310076)

C3-14 Line 317: Change to ‘dynamics’

R3-14 Does not apply anymore after the rewording.

C3-15 Line 341: not sure what is meant by ‘transect closets’??

R3-15 We believe the Reviewer means “closest transect”. We checked this and made it clear throughout the paragraph (Lines 390-393).

“To illustrate the ability but also the limitations of our method to observe shoreline variability from satellite, Figure S6 shows a comparison between the closest satellite transects and various ground measurements of some of the longest shoreline datasets around the world:...”

and Lines (406-408):

“These correlation coefficients between our satellite-derived shorelines representative of the regional scale, and in-situ local shorelines range from 0.38 to 0.61 at these sites, despite distances of up to tens of kilometers between our closest transects and the sites.”

C3-16 Line 346: Need a ‘.’ after ‘...2016’

R3-16 Done.

C3-17 Line 353: Provide some citations of this extensive validation.

R3-17 We have now added some key references (Lines 420-423) in the text that reads:

“The offshore wave energy flux, proportional to and here directly taken as $H_s 2xT_p$ where H_s is the significant wave height and T_p the swell peak period (Mentaschi, et al., 2017) was extracted from ERA5 (Hersbach et al, 2020), developed by the European Centre for Medium-Range Weather Forecasts model (ECMWF), at $0.25^\circ \times 0.25^\circ$ and hourly temporal resolution. The ERA5 reanalysis uses a coupled ocean wind-wave and atmospheric model, which has been extensively validated (Dee et al., 2011; Sterl and Caires, 2005; Caires et al., 2006).”

Newly cited Literature:

Caires, S., Swail, V. & Wang, X. Projection and analysis of extreme wave climate. J. Clim. 19, 5581–5605 (2006)

Dee, D. P. et al. The era-interim reanalysis: Configuration and performance of the data assimilation system. Q. J. royal meteorological society 137 553–597 (2011).

Sterl, A. & Caires, S. Climatology, variability and extrema of ocean waves: the web-based KNMI/ERA-40 wave atlas. Int. J. Climatol. 25, 963–977 (2005).

REVIEWERS' COMMENTS

Reviewer #1 (Remarks to the Author):

I have reviewed the response of the authors to my earlier review. I commend them on their thorough and positive response. As the result the manuscript is significantly improved and all my earlier reservations are addressed. Hence I recommend publication as resubmitted.

Reviewer #3 (Remarks to the Author):

The authors have done a great job of addressing all of the review comments. The manuscript is very well written and therefore I am not submitting an annotated version of the manuscript. The paper is now suitable for publication in Nature Communications. I have only a few relatively minor comments that should be addressed:

Lines 27-28: This phrase needs to be modified: '... waves affect both erosion/accretion and storm surge water levels, ...'. while waves do indeed affect total water levels – via wave setup and swash (runup) – they do not affect storm surges which, as the authors note on lines 127-128, is simply comprised of wind setup and surface atmospheric pressure effects.

Lines 58-62. The first sentence in this pair refers to global scale limitations. The 2nd sentence refers to applying models at regional scale. I don't see the connection. Are the authors arguing against global scale analysis? Please clarify.

Lines 70-71: studies focused on the influence of ENSO on the interannual wave climate and the subsequent influence on shorelines have existed for decades – e.g., see the numerous studies on the topic by Komar and many others.

We express our gratitude to the editorial team and the reviewers for carefully reading our revised manuscript and for providing constructive comments. We have given our full attention to the last comments made by Reviewer #3 and revised the manuscript accordingly. Reviewers' feedback and suggestions have resulted in an improved manuscript, which we hope is now suitable for publication. Our detailed responses to each comment are provided below. For clarity, we use the following color code:

reviewer comments

our response

our modifications in the manuscript

Reviewer #1 (Remarks to the Author):

C1-1 I have reviewed the response of the authors to my earlier review. I commend them on their thorough and positive response. As the result the manuscript is significantly improved and all my earlier reservations are addressed. Hence I recommend publication as resubmitted.

R1-1 We thank the Reviewer for his/her previous suggestions, encouragement, and for approving the modified version of the manuscript.

Reviewer #3 (Remarks to the Author):

C3-1 The authors have done a great job of addressing all of the review comments. The manuscript is very well written and therefore I am not submitting an annotated version of the manuscript. The paper is now suitable for publication in Nature Communications. I have only a few relatively minor comments that should be addressed:

R3-1 We thank the Reviewer for his earlier constructive feedback on our manuscript, which helped us to improve it, and we hope that it is now more suitable for publication after the slight suggested below.

C3-2 Lines 27-28: This phrase needs to be modified: '... waves affect both erosion/accretion and storm surge water levels, ...'. while waves do indeed affect total water levels – via wave setup and swash (runup) – they do not affect storm surges which, as the authors note on lines 127-128, is simply comprised of wind setup and surface atmospheric pressure effects.

R3-2 The Reviewer is indeed correct. Thank you for pointing this out. We have corrected accordingly.

New sentence (Lines 27-28) in the Abstract reads:

'While sea level directly affects coastal mobility, waves affect both erosion/accretion and total water levels, and rivers affect coastal sediment budgets and salinity-induced water levels.'

C3-3 Lines 58-62. The first sentence in this pair refers to global scale limitations. The 2nd sentence refers to applying models at regional scale. I don't see the connection. Are the authors arguing against global scale analysis? Please clarify.

R3-3 We thank the Reviewer. The wording was a bit odd, a "global" terminology was certainly missing in the last sentence of the paragraph. We have modified the text accordingly (Lines 55-59) (Please note that the references are now numbered in the manuscript, according to the editorial requirements):

“A global application of existing ocean models coupled with waves-sea level-rivers-sediment (e.g., CROCO, Marchesiello et al., 2022, Delft3D, Roelvink et al., 1995) is currently out of reach. Thus, physically simplified shoreline models (e.g., CASCADE, Abessolo et al., 2021; Larson et al., 2003, COSMOS , Vitousek et al., 2017, ShorelineS, Roelvink et al., 2020, LX-Shore, Robinet et al., 2018; Tran et al., 2021) have a bright future as potential tools for investigating drivers of shoreline change at regional to global scales.”

C3-4 Lines 70-71: studies focused on the influence of ENSO on the interannual wave climate and the subsequent influence on shorelines have existed for decades – e.g., see the numerous studies on the topic by Komar and many others..

R3-4 We appreciate the reviewer's comment. We believe that this was due to an inappropriate logical order of sentences, as we mentioned in the next paragraph a long list of previous works showing the link between ENSO and Pacific evolution. We take the opportunity of the reviewer's comment to add an earlier reference by Allan and Komar (2006) that was missing from our list. We have reformulated the text so that it reads more clearly (Lines 76-82) (Please note that the references are now numbered in the manuscript, according to the editorial requirements):

“While ENSO is known to be unambiguously dominant in the Pacific with strong impacts on the shoreline and coastal ecosystems (Storlazzi and Griggs, 1998; Ranasinghe et al., 2004; Allan and Komar, 2006; Biribo and Woodroffe, 2013; Barnard et al., 2011, 2015, 2017; Harley et al., 2017; Young et al., 2018; Cuttler et al., 2020; Duke et al., 2022; Vos et al., 2023), the possible linkages between ENSO and the key drivers of shoreline change at global scale have not yet been fully explored. In particular, the recent rejuvenation of ENSO research has led to many theoretical breakthroughs in understanding its complex and diverse regimes (Timmermann et al., 2018). On a global basis, other climate modes can also significantly modulate coastal drivers in other ocean basins.”

Another important reference has been added to the list:

“Allan, J. C., and Komar, P. D. (2006). Climate controls on US West Coast erosion processes. Journal of coastal research, 22(3), 511-529.”